# ArtiMuse: Fine-Grained Image Aesthetics Assessment with Joint Scoring and Expert-Level Understanding

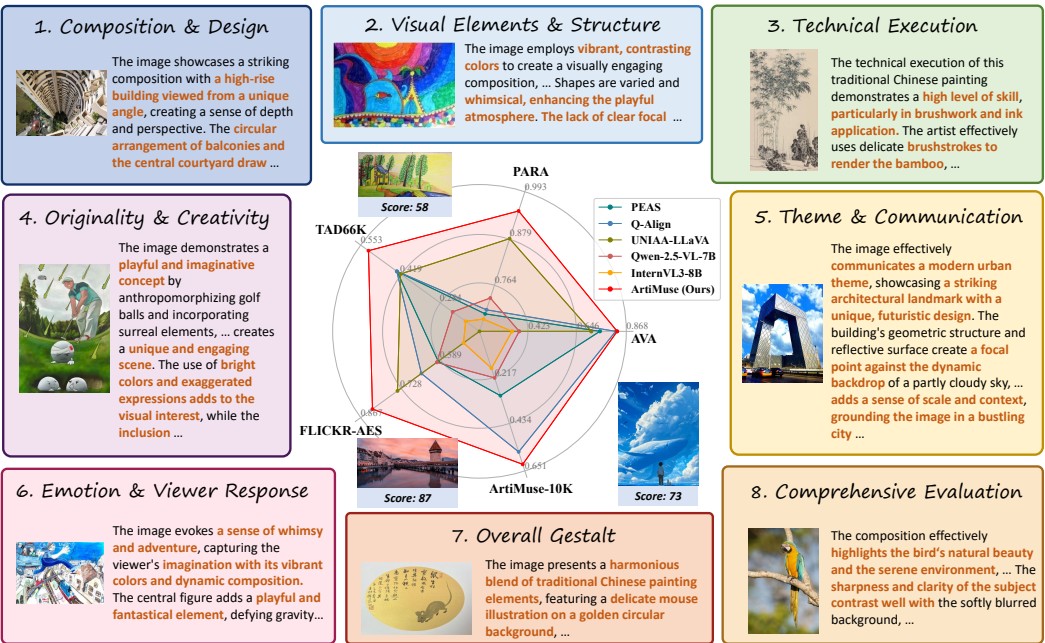

Figure 1: ArtiMuse provides granular, expert-level textual understanding results for images across eight fine-grained aesthetic attributes. Additionally, it achieves precise image aesthetics scoring, significantly outperforming state-of-the-art models across multiple widely-used benchmarks.

## Abstract

The rapid advancement of educational applications, artistic creation, and AI-generated content (AIGC) technologies has substantially increased practical requirements for comprehensive Image Aesthetics Assessment (IAA), particularly demanding methods capable of delivering both quantitative scoring and professional understanding. Multimodal Large Language Model (MLLM)-based IAA methods demonstrate stronger perceptual and generalization capabilities compared to traditional approaches, yet they suffer from modality bias (score-only or text-only) and lack fine-grained attribute decomposition, thereby failing to support further aesthetic assessment. In this paper, we present: (1) **ArtiMuse**, an innovative MLLM-based IAA model with Joint Scoring and Expert-Level Understanding capabilities; (2) **ArtiMuse-10K**, the first expert-curated image aesthetic dataset comprising 10,000 images spanning 5 main categories and 15 subcategories, each annotated by professional experts with 8-dimensional attributes analysis and a holistic score. Both the model and dataset will be made public.

## 1 Introduction

In the era of digitalization and visual information explosion, images have become an essential medium for human beings to perceive the world, document daily life, and express emotions. From professional photography and painting to casual snapshots and sharing, images play a crucial role in conveying aesthetic values, emotional narratives, and storytelling. The advent of artificial intelligence generated content (AIGC) technologies dreamlike.art (2023); Labs (2024); Rombach et al.

(2021) has further democratized visual content creation. However, this abundance of visual content also poses new challenges for quality assessment, filtering, and recommendation. While existing image quality assessment (IQA) techniques You et al. (2024b;a; 2025) have matured in detecting low-level degradations such as blurriness, noise, and compression artifacts, they largely focus on the technical fidelity of images and fail to capture their higher-level aesthetic attributes. Image aesthetics assessment (IAA) Huang et al. (2024); Gao et al. (2024); Jin et al. (2024), which evaluates aspects such as artistic appeal, color harmony, and emotional expression, is increasingly recognized as a fundamental capability in applications including AIGC content evaluation, creative assistance, and photography education.

Despite the growing demand, current IAA methods face notable limitations. Most existing approaches rely on simplistic score predictions without capturing the inherent subjectivity, multidimensionality, and nuanced interpretations of aesthetics. Moreover, available datasets are often small in scale, coarse in granularity, and lack professionally curated annotations based on established aesthetic theories. This gap severely limits the ability of state-of-the-art multimodal large models (MLLMs) Zhu et al. (2025); Bai et al. (2025) to understand and reason about aesthetics.

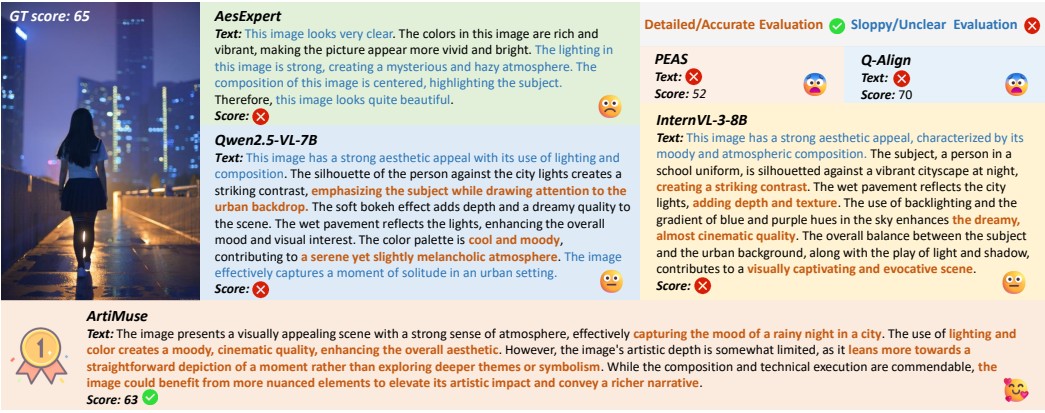

Figure 2: In comparison with existing models, ArtiMuse outperforms them by simultaneously achieving both accurate evaluation and precise aesthetics scoring in multi-dimensional assessments.

To address these challenges, we introduce **ArtiMuse**, a multimodal large language model (MLLM) for professional aesthetic understanding, together with **ArtiMuse-10K**, a meticulously curated, expert-annotated dataset. Collaborating with domain experts in aesthetics, each with 3 to over 30 years of experience, we systematically define eight explainable and fine-grained aesthetic attributes, covering aspects such as Composition & Design, Visual Elements & Structure, and Originality & Creativity, among others. Based on these attributes, we construct ArtiMuse-10K, the largest and most comprehensive fine-grained image aesthetics dataset to date, featuring both quantitative aesthetic scores and expert-written textual analyses across diverse visual domains, including graphic design, 3D design, AIGC-generated images, photography, and painting & calligraphy.

Leveraging this dataset, ArtiMuse jointly predict aesthetic scores and generate expert-level, fine-grained textual analysis, advancing aesthetic AI from mere score prediction toward holistic, interpretable reasoning. Notably, ArtiMuse achieves state-of-the-art performance across multiple widely used aesthetics benchmarks, demonstrating its robust generalization ability and superior performance in both quantitative assessment and qualitative explanation, as shown in Fig. 1 and Fig. 2.

In addition, a core technical challenge in aesthetics modeling lies in continuous score prediction using MLLMs, which are inherently designed for discrete token generation. Existing methods such as Q-Align Wu et al. (2024b) attempt to transform continuous scores into discrete ratings, and then reconstruct continuous values by weighted averaging over rating logits. However, this discretization inevitably incurs significant information loss and often leads to inaccurate predictions. To overcome this limitation, we propose a novel *Token As Score* strategy that densely maps predefined discrete tokens to continuous values. Specifically, we utilize existing tokens within the native LLM tokenizer to represent numeric values, thus eliminating the need to expand the vocabulary or retrain the tokenizer. This lightweight yet effective technique enables precise and robust modeling of continuous values within the MLLM framework, substantially improving the fidelity of aesthetics scoring.

Our main contributions can be summarized as follows:

**(1) ArtiMuse-10K**, a comprehensive and meticulously annotated image aesthetic assessment dataset containing 10,000 images spanning over 5 main categories and 15 subcategories. Each image is manually annotated by professional experts with detailed textual evaluations across 8 aesthetic attributes, accompanied by an overall aesthetics score. As far as we know, This dataset represents the most extensive expert-curated resource for aesthetics assessment to date.

**(2) ArtiMuse**, a novel image aesthetics assessment model, is capable of performing fine-grained expert-level textual analysis and providing accurate aesthetic scores. ArtMuse exhibits significantly superior aesthetic assessment expertise and fine-grained analysis compared to other IAA models and general-purpose MLLMs.

**(3) Token As Score**, which enables precise continuous aesthetics scoring in MLLMs by mapping existing tokens to numeric values, avoiding quantization loss and tokenizer changes. It offers a lightweight, effective solution for accurate and stable score prediction.

## 2 RELATED WORK

### 2.1 MULTI-MODALITY LARGE LANGUAGE MODELS

With the advancement of MLLMs Achiam et al. (2023); Team et al. (2023); Bai et al. (2025); Zhu et al. (2025), their ability has expanded from basic image-text matching to understanding high-level semantic content, offering new possibilities for image aesthetics assessment. However, current MLLMs still struggle with objective evaluation, often producing overly positive and superficial judgments. Moreover, the text they generate differs significantly from the professional descriptions used by human experts, making them less suitable for high-quality automated aesthetic evaluation. Therefore, systematic fine-tuning is required to optimize and guide these models.

### 2.2 IMAGE AESTHETICS ASSESSMENT

**Datasets.** As summarized in Table 1, existing IAA datasets suffer from three key limitations: (1) Many He et al. (2022); Kong et al. (2016); Yang et al. (2022); Murray et al. (2012) offer overall aesthetics scores but lack detailed evaluative descriptions, while others Achlioptas et al. (2021); Kruk et al. (2023) provide only vague comments without numerical ratings; (2) Most Jin et al. (2024); Nieto et al. (2022) focus solely on overall impressions, lacking fine-grained aesthetic attribute annotations; (3) In terms of content, datasets Kong et al. (2016); Yang et al. (2022); Murray et al. (2012) are mainly photographic, with limited inclusion of artworks Jin et al. (2024); Achlioptas et al. (2021); Yi et al. (2023) and little to no AIGC or everyday scene coverage. These gaps hinder aesthetic modeling, highlighting the need for a more diverse, well-annotated benchmark.

**Models.** IAA models have evolved from simple regression to multimodal generative evaluation with integrated language understanding. Existing approaches fall into two categories: (1) Regression-based models (e.g., TANet He et al. (2022), AesMamba Gao et al. (2024)) directly predict aesthetics scores from image features but lack interpretability and generalization; (2) MLLM-based generative models leverage vision-language understanding to align better with human perception. Instruction-tuned models Wu et al. (2024a); Yun & Choo (2024b) improve text generation but with limited granularity. AesExpert Huang et al. (2024) produces expert-style descriptions but lacks score prediction. Q-Align Wu et al. (2024b) and UNIAA Zhou et al. (2024) combine text and discrete scores, yet lack fine-grained dimension-level evaluation. To overcome these gaps, we introduce ArtiMuse, a unified model that generates expert-level analysis and accurate aesthetics scores.

## 3 ARTIMUSE-10K DATASET

### 3.1 DATASET OVERVIEW

As shown in Tab. 1, ArtiMuse-10K far exceeds existing IAA datasets in diversity and granularity. It contains 10,000 images across 5 main categories (Design, AIGC, photography, etc.) with 15 fine-grained subcategories. Each image is annotated by professional experts on eight aesthetic attributes and an overall score, offering superior professional rigor and annotation granularity.

### 3.2 IMAGE COLLECTION

Previous studies Jin et al. (2024); Wu et al. (2024b); Huang et al. (2024) have emphasized the importance of ensuring dataset diversity and extending domain coverage to enhance the quality and ro-

bustness of aesthetic assessment models. Building upon these insights, we construct ArtiMuse-10K, a high-quality dataset comprising 10,000 carefully curated images spanning five primary categories: Graphic Design, 3D Design, AIGC-generated images, Photography, and Painting & Calligraphy. These categories are subdivided into 15 distinct subcategories, such as Chinese Painting, Sculpture, and Daily Photography, ensuring comprehensive representation of diverse artistic expressions. The internal data samples and overall dataset composition are illustrated in Fig. 3 and Fig. 4, respectively.

Table 1: A Comparison between ArtiMuse-10K dataset and existing IAA datasets.

| Dataset | Main Categories | Subcategories | # Image | Score | Text Caption | # Attribute | Attribute Categories | Annotators |
|---|---|---|---|---|---|---|---|---|
| AVA Murray et al. (2012) | Photography | – | 255,528 | ✓ | ✗ | – | – | Non-Experts |
| AADB Kong et al. (2016) | Photography | – | 10,000 | ✓ | ✗ | – | – | Non-Experts |
| FLICKR-AES Ren et al. (2017) | Photography | 9 Categories | 40,499 | ✓ | ✗ | – | – | Non-Experts |
| SPAQ Fang et al. (2020) | Photography | – | 111,125 | ✓ | ✗ | – | – | Non-Experts |
| KonIQ-10K Hosu et al. (2020) | Photography | – | 10,073 | ✓ | ✗ | – | – | Non-Experts |
| ArtEmis Achlioptas et al. (2021) | Painting | – | 81,446 | ✗ | ✓ | 1 Attribute | Emotional Analysis | Non-Experts |
| RPCD Nieto et al. (2022) | Photography | – | 73,965 | ✓ | ✓ | 1 Attribute | Overall Comment | Non-Experts |
| PARA Yang et al. (2022) | Photography | – | 31,229 | ✓ | ✗ | – | – | Non-Experts |
| TAD66K He et al. (2022) | Painting, Photography | – | 66,000 | ✓ | ✗ | – | – | Non-Experts |
| Impressions Kruk et al. (2023) | Photography | – | 1,440 | ✗ | ✓ | 3 Attributes | Description, Perception, Evaluation | Non-Experts |
| BAID Yi et al. (2023) | Painting | – | 60,337 | ✓ | ✗ | – | – | Non-Experts |
| APDDv2 Jin et al. (2024) | Painting | 3 Categories | 10,023 | ✓ | ✓ | 1 Attribute | Overall Comment | Professional Experts |
| **ArtiMuse-10K (Ours)** | **Graphic Design, 3D Design, AIGC, Photography, Painting & Calligraphy** | **15 Detailed Categories** | **10,000** | ✓ | ✓ | **8 Attributes** | **Fine-grained Attributes (Composition & Design, Technical Execution, etc.)** | **Professional Experts** |

**Non-AIGC Images.** For non-AIGC images, we collaborate with domain experts to curate professionally created artworks sourced from academic settings, including student assignments and competition entries. To ensure the dataset reflects contemporary trends, we also collect a wide range of artistic and photographic works from reputable online art and photography platforms.

**AIGC Images.** We utilize state-of-the-art generative models (Stable Diffusion series Rombach et al. (2021), Dreamlike Photoreal 2.0 dreamlike.art (2023), FLUX Labs (2024), etc.) to systematically produce synthetic images. We further augment this core dataset with open-source community contributions produced using comparable architectures.

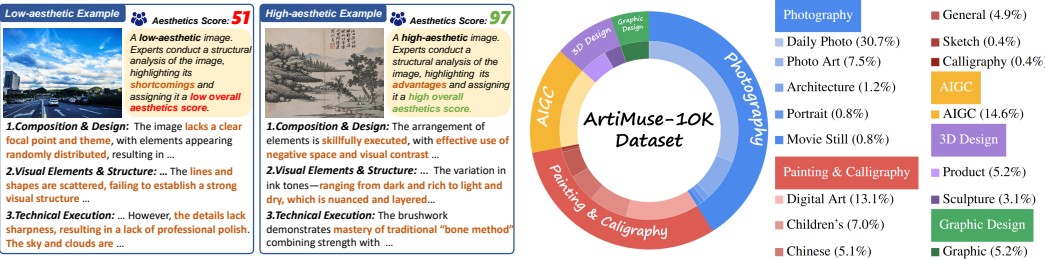

Figure 3: Data examples in ArtiMuse-10K.  Figure 4: Composition of ArtiMuse-10K.

### 3.3 AESTHETIC ATTRIBUTES

To establish a fine-grained annotated dataset for image aesthetics assessment, the primary task involves developing a comprehensive assessment system. Through systematic consultations with artistic experts, we have formulated a novel aesthetic assessment system. This system comprises 8 specific aesthetic attributes and an overall aesthetics score, systematically defining key dimensions of image aesthetics including Composition & Design, Visual Elements & Structure, Technical Execution, Originality & Creativity, Theme & Communication, Emotion & Viewer Response, Overall Gestalt and Comprehensive Evaluation. Notably, our system is content-agnostic and universally applicable to image types from natural to AIGC.

### 3.4 HUMAN ANNOTATIONS

Based on the predefined aesthetic attributes, we invite professional experts to meticulously annotate images in the ArtiMuse-10K dataset. We collaborate with domain experts whose professional experience spans a broad spectrum, ranging from at least three years to over three decades, including distinguished authorities in the field. The entire annotation process is illustrated in Fig. 5 as Type 3: Professionally Selected Images. Each image in ArtiMuse-10K is ultimately annotated with textual analysis describing eight distinct aesthetic attributes and an overall aesthetics score. Our comprehensive annotation framework enhances dataset quality and model performance by integrating multi-dimensional aesthetic attributes for fine-grained visual analysis, expert-curated scores for reliable aesthetic assessment, and rich semantic annotations for improving training robustness.

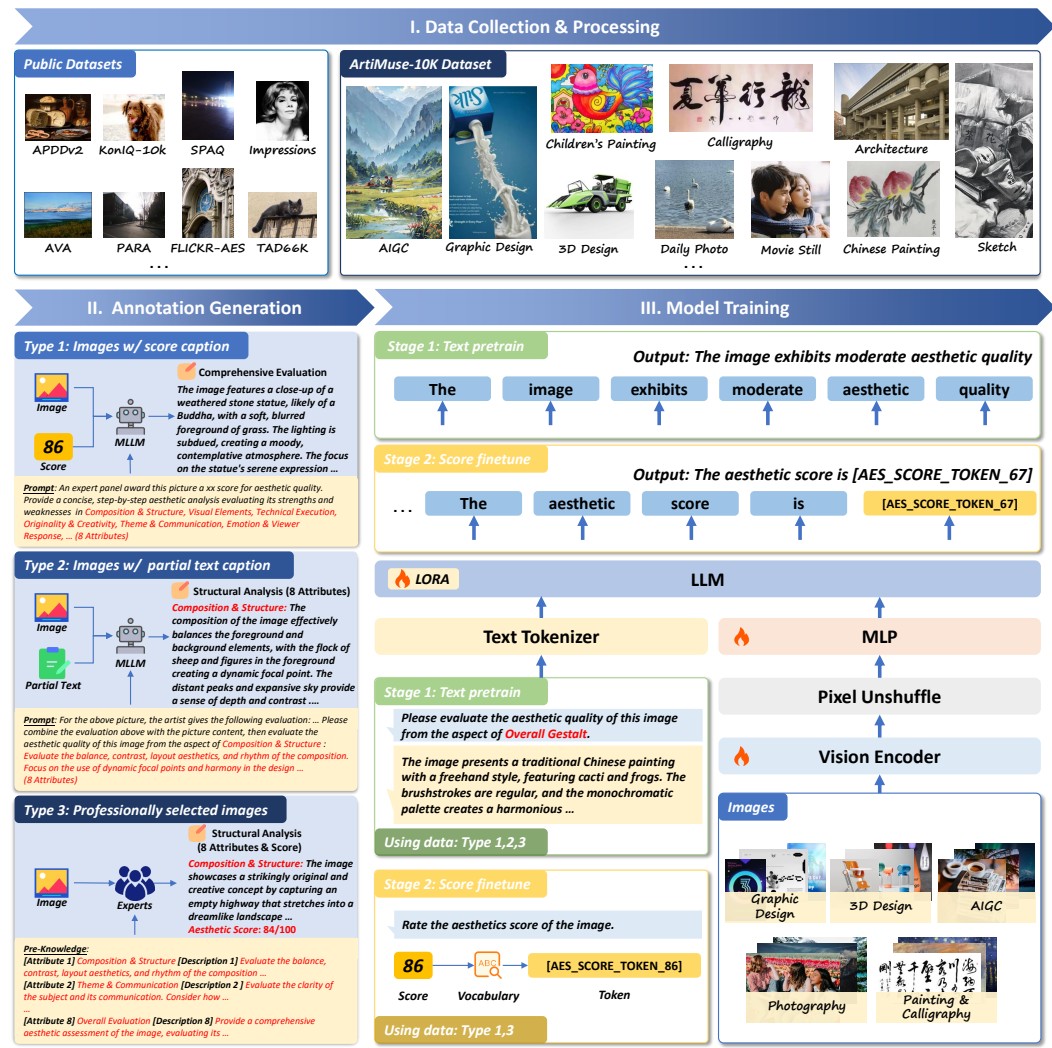

Figure 5: Overview of ArtiMuse. ArtiMuse encompasses a multi-stage pipeline spanning data collection & processing, annotation generation, and model training, systematically enhancing its text evaluation capabilities and score assessment proficiency across multiple dimensions.

## 4 METHODOLOGY

### 4.1 DATASET COLLECTION & PROCESSING

Richer data sources and more meticulous manual annotations are crucial for enhancing dataset quality. In addition to the ArtiMuse-10K dataset, we carefully curate over 350,000 high-quality annotated images from existing datasets, including APDDv2 Jin et al. (2024), PARA Yang et al. (2022), Impressions Kruk et al. (2023) and so on.

**Aesthetic Caption Quality.** We place particular emphasis on the aesthetic caption quality. Our selection criteria prioritize datasets that include valuable aesthetic-related captions such as aesthetics scores, comprehensive textual analyses, and aesthetic attribute tags. These captions are subsequently utilized in the annotation generation phase to enhance dataset quality. Ultimately, the collected captions come from annotators ranging from domain experts with **3–30 years** of professional experience to community enthusiasts and everyday users, ensuring diversity in perspectives and judgments.

**Aesthetic Quality Diversity.** Our collection specifically incorporates images with varying aesthetic qualities, including intentionally retained lower-quality samples, to address both dataset diversity requirements and mitigate the prevalent preference bias observed in contemporary LLMs. This carefully balanced composition strategy enhances model training through controlled inclusion of suboptimal visual materials, thereby improving discriminative capabilities in aesthetic assessment.

## 4.2 ANNOTATION GENERATION

The Annotation Generation stage aims to enrich the dataset with detailed descriptive and evaluative annotations, illustrated in Fig. 5. This process involves creating distinct annotation types based on the available information for each image. **Type 1:** For images with only score caption, we leverage this global quality assessment to generate holistic analyses. We design a prompt to guide the MLLM in producing a comprehensive evaluation based on predefined aesthetic attributes, while incorporating both the score and visual input. **Type 2:** For images with partial text captions containing specific aesthetic descriptions, we employ a prompt to instruct the MLLM to generate fine-grained evaluations. For each image, the model produces a structural analysis across 8 aesthetic attributes, utilizing both the textual and visual inputs. **Type 3:** For professionally selected images, we engage experts to conduct structural analysis based on pre-defined aesthetic attributes, along with providing an overall aesthetics score. More details are in the Appendix B.

**Importance of Manual Annotations.** Although MLLMs demonstrate strong aesthetic evaluation capabilities, our empirical analysis reveals a systematic bias: they tend to generate overwhelmingly positive assessments regardless of the actual image quality, as shown in Fig. 7. This positivity bias leads to annotations that poorly reflect true aesthetic merit. To address this limitation, we incorporate professional human evaluations to provide balanced and reliable ground-truth annotations.

## 4.3 TRAINING STRATEGY

ArtiMuse is built on InternVL-3-8B Zhu et al. (2025). We modify the dynamic resolution strategy to a fixed-resolution approach while retaining the remaining components. The training process consists of two distinct phases: text pretraining and score fine-tuning, as illustrated in Fig. 5. In both stages, we jointly train the vision encoder, MLP, and LLM components, with the LLM undergoing LoRA-based fine-tuning. The ArtiMuse uses common GPT loss Radford et al. (2019), i.e. minimizing the cross-entropy loss between the predicted logits and target tokens.

**Text Pretrain.** The text pretraining phase utilizes our complete collected image dataset, where each image is paired with its corresponding aesthetic analysis caption generated during the annotation generation stage. This phase aims to equip the model with accurate structural aesthetic analysis capabilities while largely preserving the MLLM's pretrained knowledge. To achieve this balance, we apply LoRA fine-tuning specifically to the LLM component.

**Score Finetune.** After establishing foundational aesthetic understanding through pretraining, we proceed to score fine-tuning. In this phase, we convert each image's overall aesthetics score into a specialized scoring token designed exclusively for aesthetics scoring, which then serves as the training caption. Inspired by previous works Wu et al. (2024b); Zhou et al. (2024); Li et al. (2025), we propose a novel score prediction strategy called *Token As Score*, which eliminates the need for vocabulary expansion or tokenizer retraining. Specifically, we designate 101 existing tokens as `[Aes_Score_Token]`s, each corresponding to integer scores ranging from 0 to 100. We select tokens that are concise and inherently carry ordinal semantic information from the vocabulary. In our implementation, we employ twin-letter combinations as tokens (e.g., Score 1 is represented as `[Aes_Score_Token_1]`, where the actual token is `ab`. See Appendix for more details). During data preprocessing, we first normalize aesthetics scores to the [0,100] range and then map them to their corresponding tokens. This methodology enables the construction of training data where continuous scores are discretized into token representations. The model is subsequently fine-tuned to predict these discrete tokens. During inference, we convert the predicted tokens back to their numerical values, and the final aesthetics score is derived by computing the expectation over the probability distribution of all possible score tokens. Specifically, we denote $l_i$ and $p_i$ for logits and probability of `[Aes_Score_Token_i]`, the final aesthetics score $S_{\text{Aes}}$ is compute as:

$$S_{\text{Aes}} = \sum_{i=0}^{100} i \times p_i = \sum_{i=0}^{100} i \times \frac{e^{l_i}}{\sum_{j=0}^{100} e^{l_j}} \tag{1}$$

**Why Token As Score?** Current approaches for scoring with MLLMs primarily fall into two categories: (1) directly prompting the LLM to output scores as text (*Text As Score*), or (2) predefining discrete levels corresponding to specific score intervals and computing the final score based on the model's predicted token distribution (*Level As Score*). Previous works Wu et al. (2024b); Li et al. (2025); Zhou et al. (2024) demonstrate that directly generating scores as text leads to severe hallucination issues. Thus, we adopt the Token As Score approach and investigate the impact of token granularity on model performance. Fig. 6 presents a comparison of these score prediction methods.

Figure 6: Comparison of score prediction methods. Token As Score features a more rational design and delivers more precise results.

**Maintaining Text Ability.** A widely recognized challenge in IAA and IQA tasks is that MLLMs often struggle to simultaneously preserve their textual understanding and scoring capabilities You et al. (2024b; 2025). Since the training data in the score fine-tuning phase is significantly more monotonous than in text pretraining, full fine-tuning of the LLM can easily degrade its structural aesthetic analysis ability. To mitigate this issue while maintaining proficiency, we employ LoRA-based fine-tuning for the LLM, enabling the model to retain both linguistic and scoring capabilities.

## 5 EXPERIMENTS

### 5.1 IMPLEMENTATION DETAILS

In our experiments, we adopt InternVL-3-8B Zhu et al. (2025) as the base model initialized with its pretrained weights. During text pretraining, we implement a batch size of 128 and learning rate of $4e - 5$ with a cosine annealing schedule Loshchilov & Hutter (2017), training for one epoch to balance convergence with prior knowledge preservation. For the score fine-tuning, we maintain the batch size at 128 while adjusting the learning rate to $2e - 5$ across 2 training epochs. We maintain identical configurations across all experiments, with all training conducted on 4 * NVIDIA A100 80GB GPUs. Text pretraining completes within 5 hours, while score fine-tuning converges efficiently in just 10 minutes on ArtiMuse-10K and 4 hours on AVA (2M-scale dataset).

### 5.2 STRUCTURAL AESTHETIC ANALYSIS

**Judging by MLLM.** To evaluate current models' ability of structural aesthetic analysis, we design a judgement framework leveraging the superior comprehension power of MLLM. An image is presented to both experts and various models to generate aesthetic analysis on 8 aesthetics attributes. Then a judging MLLM selects which model performs best across each attribute, using the human expert's description as a reference. The results in Tab. 2 show ArtiMuse outperforms other models across 8 aesthetic attributes, demonstrating superior structural aesthetic analysis capability.

**Judging by Human.** In addition, we conduct a user study where participants are asked to compare and vote for the model they perceive as producing higher-quality aesthetic analysis. The proportion of selections for each model, presented as Human Rate in Table 2, demonstrates that our approach achieves a significantly higher preference rate compared to other methods.

**Qualitative Comparison.** Fig. 7 presents a systematic evaluation of aesthetic analysis performance across different models. Our approach demonstrates consistent superiority in analyzing both natural and AIGC images, with particular strengths in identifying key aesthetic elements such as compositional cohesion and characteristic AIGC artifacts. More results are provided in the Appendix E.

### 5.3 AESTHETICS SCORING

**Comparison across Multiple Image Aesthetics Scoring Datasets.** We evaluate the performance of ArtiMuse against other models across multiple Image aesthetics scoring datasets. For models unable for test (TANet He et al. (2022), AesMamba Gao et al. (2024), UNIAA-LLaVA Zhou et al. (2024), Next Token Is Enough Li et al. (2025)), we directly adopt the test results reported in their original papers. For models with unclear training protocols or those trained on general scenarios (MUSIQ Ke et al. (2021), VILA Lin et al. (2023), mPLUG-Owl2 Ye et al. (2023), ShareGPT-4V Chen et al. (2024), Qwen-2.5-VL-7B Bai et al. (2025), InternVL3-8B Zhu et al. (2025), Q-Instruct Wu et al. (2023), PEAS Yun & Choo (2024a)), we test their official released models. Both Q-Align Wu et al. (2024b) and our proposed model are fine-tuned on each target dataset. As shown in Tab. 3, ArtiMuse demonstrates superior performance, achieving nearly the highest metrics across all datasets. Notably, it outperforms other models by over 0.05 PLCC on the PARA Yang et al. (2022) and ArtiMuse-10K datasets, demonstrating its accurate aesthetics scoring capability.

Table 2: The selection rates of different models. For the first 8 aesthetic attributes, evaluations are performed by Gemini-2.0-flash, while Human Rate is provided by volunteer participants.

| Aesthetic Attributes | AesExpert Huang et al. (2024) | Qwen-2.5-VL-7B Bai et al. (2025) | InternVL3-8B Zhu et al. (2025) | ArtiMuse |
|---|---|---|---|---|
| Composition & Design | 0.0% | 12.7% | 10.4% | **76.9%** |
| Visual Elements & Structure | 0.0% | 19.3% | 16.5% | **64.2%** |
| Technical Execution | 0.0% | 9.9% | 10.4% | **79.7%** |
| Originality & Creativity | 0.0% | 13.7% | 8.5% | **77.8%** |
| Theme & Communication | 0.9% | 17.5% | 24.1% | **58.5%** |
| Emotion & Viewer Response | 0.0% | 17.5% | 24.1% | **58.5%** |
| Overall Gestalt | 0.0% | 14.6% | 9.4% | **75.9%** |
| Comprehensive Evaluation | 0.0% | 17.5% | 10.8% | **71.7%** |
| **Attributes Average** | 0.1% | 14.3% | 14.5% | **71.1%** |
| **Human Rate** | 1.5% | 11.5% | 19.2% | **67.8%** |

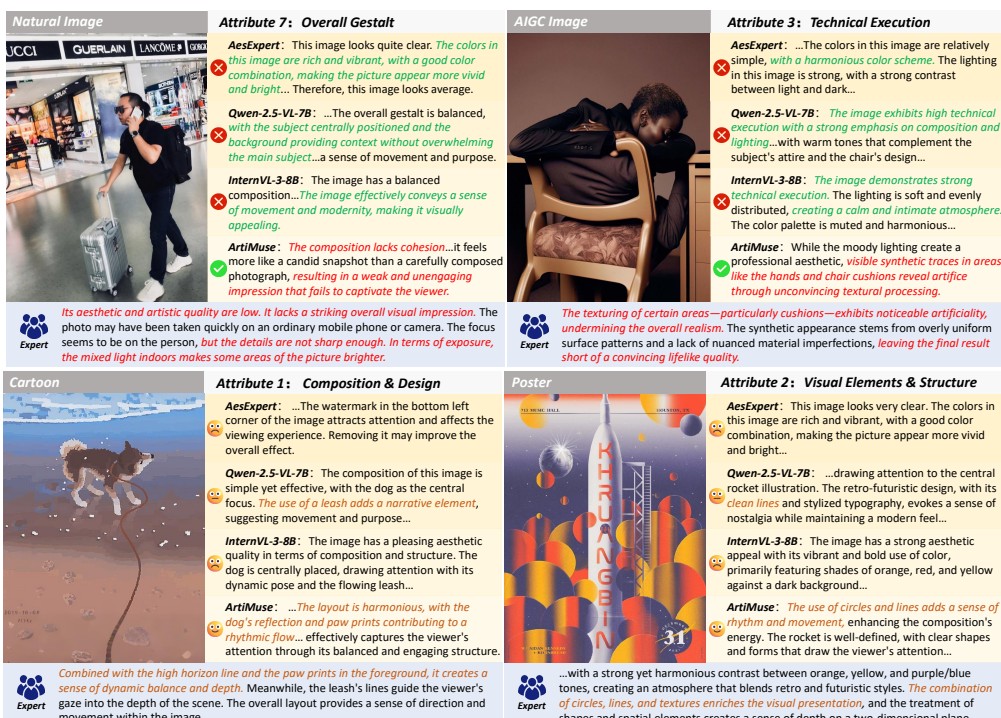

Figure 7: Structural aesthetic analysis results. Red, green, and brown denote positive, negative, and expert-level analyses, respectively. ArtiMuse uniquely identifies flaws in low-aesthetic images while providing professional assessment of high-aesthetic images, capabilities absent in other models.

**Generalization Ability.** We compare the generalization capabilities of ArtiMuse and Q-Align, the top-performing baseline model in comparison. Both models are fine-tuned solely on the largest AVA dataset Murray et al. (2012) and subsequently evaluated on out-of-distribution datasets without additional adaptation. As presented in Table 3, ArtiMuse consistently achieves superior performance over Q-Align across all benchmark datasets. Remarkably, ArtiMuse's zero-shot transfer performance exceeds that of specialized IAA models, highlighting its exceptional generalization ability.

**Discussion of Image Aesthetics Scoring Datasets.** Prior work Wu et al. (2024b); You et al. (2025); Zhou et al. (2024); Li et al. (2025) has consistently demonstrated that IAA remains a challenging task due to the subjective nature of aesthetic perception and the substantial distributional shifts across different datasets. Our results in Tab. 3 further corroborate this observation: while models can achieve strong performance when fine-tuned on a single dataset, their accuracy often degrades significantly when evaluated on unseen datasets.

### 5.4 ABLATION STUDIES

**Datasets Variants.** We conduct 4 experiments (a)-(c) to systematically validate the contribution of each dataset component, as shown in Tab. 4. The results demonstrate consistent performance drop when any component is removed, with the most significant drop occurring upon exclusion of the Images w/ score caption subset, for the subset's inclusion of data from AVA. The results underscore the critical impact of dataset composition on model performance.

Table 3: Comparison on aesthetics scoring. The best and second-best performances are highlighted in red and blue, respectively. † Results are taken directly from original papers as these models cannot be tested. * Results are trained only on AVA to compare the generalization ability. For models without scoring capability, we prompt them to directly output scores as text for evaluation.

| Model | AVA Murray et al. (2012) | | PARA Yang et al. (2022) | | TAD66K He et al. (2022) | | FLICKR-AES Ren et al. (2017) | | ArtiMuse-10K | |
|---|---|---|---|---|---|---|---|---|---|---|
| | SRCC | PLCC | SRCC | PLCC | SRCC | PLCC | SRCC | PLCC | SRCC | PLCC |
| *Traditional Models* | | | | | | | | | | |
| MUSIQ Ke et al. (2021) | 0.818 | 0.819 | 0.488 | 0.461 | 0.407 | 0.434 | 0.533 | 0.569 | 0.249 | 0.230 |
| TANet He et al. (2022) † | 0.758 | 0.765 | – | – | 0.513 | 0.531 | – | – | – | – |
| VILA Lin et al. (2023) | 0.776 | 0.775 | 0.651 | 0.658 | 0.418 | 0.444 | 0.616 | 0.645 | 0.273 | 0.268 |
| AesMamba Gao et al. (2024) † | 0.774 | 0.769 | 0.936 | 0.902 | 0.511 | 0.483 | – | – | – | – |
| *MLLMs for General-Purpose Applications* | | | | | | | | | | |
| mPLUG-Owl2 Ye et al. (2023) | 0.206 | 0.211 | 0.376 | 0.372 | 0.089 | 0.106 | 0.382 | 0.359 | 0.159 | 0.145 |
| ShareGPT-4V Chen et al. (2024) | 0.213 | 0.199 | 0.509 | 0.417 | 0.097 | 0.091 | 0.335 | 0.289 | 0.076 | 0.057 |
| Qwen-2.5-VL-7B Bai et al. (2025) | 0.391 | 0.371 | 0.721 | 0.743 | 0.240 | 0.242 | 0.621 | 0.578 | 0.256 | 0.179 |
| InternVL3-8B Zhu et al. (2025) | 0.364 | 0.332 | 0.667 | 0.693 | 0.191 | 0.203 | 0.553 | 0.459 | 0.187 | 0.157 |
| *MLLMs for Image Aesthetics Assessment* | | | | | | | | | | |
| Q-Instruct Wu et al. (2023) | 0.318 | 0.338 | 0.569 | 0.724 | 0.122 | 0.159 | 0.259 | 0.299 | -0.045 | -0.056 |
| PEAS Yun & Choo (2024a) | 0.748 | 0.748 | 0.686 | 0.700 | 0.415 | 0.444 | 0.577 | 0.613 | 0.306 | 0.293 |
| Q-Align Wu et al. (2024b) | 0.822 | 0.817 | 0.913 | 0.888 | 0.501 | 0.531 | 0.798 | 0.818 | 0.551 | 0.573 |
| UNIAA-LLaVA Zhou et al. (2024) † | 0.713 | 0.704 | 0.864 | 0.895 | 0.411 | 0.425 | 0.724 | 0.751 | – | – |
| Next Token Is Enough Li et al. (2025) † | 0.828 | 0.825 | – | – | 0.413 | 0.444 | – | – | – | – |
| **ArtiMuse (Ours)** | 0.827 | 0.826 | 0.936 | 0.958 | 0.510 | 0.543 | 0.814 | 0.837 | 0.614 | 0.627 |
| *Comparison of Generalization Ability* | | | | | | | | | | |
| Q-Align * | 0.822 | 0.817 | 0.694 | 0.711 | 0.417 | 0.445 | 0.643 | 0.664 | 0.337 | 0.320 |
| ArtiMuse (Ours) * | 0.827 | 0.826 | 0.697 | 0.725 | 0.419 | 0.451 | 0.647 | 0.676 | 0.395 | 0.376 |

**Training Strategy.** Comparative analysis between (d) and (h) reveals that full fine-tuning significantly impacts model performance, primarily due to the loss of fundamental aesthetic priors acquired during the text pretraining phase. This finding is further substantiated by the comparison between (e) and (h), which conclusively demonstrates the effectiveness of our proposed 2-stage training paradigm. The results indicate that preserving pretrained text representations while adapting to score prediction tasks yields superior performance compared to end-to-end joint training approaches.

**Score Prediction.** We systematically examine score prediction strategies in (f)–(h). Compared with (h), (f) demonstrates suboptimal performance, confirming that Text As Score suffers from severe hallucination issues. Exp. (g) adopts Level As Score from Q-Align but performs worse than (h), as predicting with only 5 discrete levels lacks granularity and the level words can be further decomposed into tokens, introducing noise. Interestingly, (f) is comparable to (g), highlighting the importance and effectiveness of text pretraining in our pipeline. We further investigate Token As Score strategies (Fig. 8), observing that too few tokens lack sufficient granularity, whereas too many introduce excessive complexity. Moreover, expanding tokens (newly added to the vocabulary) or using non-ordered tokens consistently degrades performance. Guided by these findings, we adopt 100 existing tokens as the final setting, with further details provided in the Appendix C.

Table 4: Ablation studies. The table compares different combinations of dataset variants, training, and training methods, with evaluation metrics SRCC and PLCC reported for AVA dataset.

| Exp. | Images w/ Score Caption | Images w/ Partial Text Caption | Professionally Selected Images | Training Strategies | Score Prediction | SRCC | PLCC |
|---|---|---|---|---|---|---|---|
| (a) | ✓ | ✓ | – | LLM LoRA / 2-Stage Training | Token As Score (100 Tokens) | 0.824 | 0.825 |
| (b) | – | ✓ | ✓ | LLM LoRA / 2-Stage Training | Token As Score (100 Tokens) | 0.621 | 0.627 |
| (c) | ✓ | – | ✓ | LLM LoRA / 2-Stage Training | Token As Score (100 Tokens) | 0.825 | 0.824 |
| (d) | ✓ | ✓ | ✓ | LLM Full-finetune / 2-Stage Training | Token As Score (100 Tokens) | 0.816 | 0.814 |
| (e) | ✓ | ✓ | ✓ | LLM LoRA / Joint Training | Token As Score (100 Tokens) | 0.821 | 0.820 |
| (f) | ✓ | ✓ | ✓ | LLM LoRA / 2-Stage Training | Text As Score | 0.820 | 0.819 |
| (g) | ✓ | ✓ | ✓ | LLM LoRA / 2-Stage Training | Level-As-Score (Q-Align) | 0.820 | 0.818 |
| (h) | ✓ | ✓ | ✓ | LLM LoRA / 2-Stage Training | Token As Score (100 Tokens) | **0.827** | **0.826** |

Figure 8: Explorations of Token As Score strategies on AVA and ArtiMuse-10K datasets. Token denotes the tokens mapping to aesthetic scores in ArtiMuse.

## 6 CONCLUSION

We introduce ArtiMuse-10K, a large expert-annotated dataset for image aesthetics assessment (IAA), and ArtiMuse, the first model to achieve expert-level textual evaluation and precise aesthetics scoring. We further propose Token As Score, a lightweight yet effective method for continuous score prediction in MLLMs. Together these contributions will advance the field of IAA by providing more comprehensive dataset, more superior model, and more efficient scoring paradigm.

**Limitations.** The current model is limited to understanding and analyzing, and is unable to provide professional aesthetic enhancement recommendations, which will be addressed in future work.

ETHICS STATEMENT

In dataset collection, we ensured that all image sources comply with ethical standards and originate from a wide range of domains, thereby providing rich diversity and minimizing potential personal bias. For annotations, we engaged annotators ranging from domain experts with 3–30 years of professional experience to community enthusiasts and everyday users, thereby ensuring diversity in perspectives and judgments. This combination of ethically sourced data and heterogeneous annotators helps mitigate bias and contributes to a fair and inclusive dataset. More details are provided in Appendix A.

REPRODUCIBILITY STATEMENT

We provide detailed reproducibility guidelines to ensure that our work can be reliably replicated. Specifically, we include comprehensive descriptions of the experimental settings, the Token-As-Score strategy, and other implementation details in the Appendix D. Furthermore, we provide the corresponding code and scripts in the supplementary materials.

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

# A ARTIMUSE-10K DATASET DETAILS

## A.1 DETAILS OF AESTHETIC ATTRIBUTES

The ArtiMuse-10K dataset employs structural analysis for textual annotations, with each image evaluated across eight fine-grained aesthetic attributes. These attributes were rigorously defined by a panel of domain experts, all of whom possess at least **3 years** of formal training in aesthetics, with the most senior member boasting **over 30 years** of professional experience in the field. This ensures comprehensive coverage of key image aesthetics dimensions while maintaining robust generalizability across diverse image types—including designs, photographs, paintings, calligraphy, and AI-generated content (AIGC) images. The detailed of these attributes are presented in Tab. A.1.

Table 5: Aesthetic attributes and their descriptions of ArtiMuse-10K dataset.

| No. | Attribute | Description |
|---|---|---|
| 1 | Composition & Design | Evaluate the balance, contrast, layout aesthetics, and rhythm of the composition. Focus on the use of dynamic focal points, unity, and harmony in the design. |
| 2 | Visual Elements & Structure | Analyze the interplay of color, geometry, spatial organization, and illumination to optimize visual contrast and structural clarity. |
| 3 | Technical Execution | Examine the mastery of medium and materials, including brushstrokes, focus, exposure, light handling, as well as clarity and resolution of the image. |
| 4 | Originality & Creativity | Analyze the uniqueness of the concept and execution, focusing on how the work exceeds common styles with imagination, and creative breakthroughs. |
| 5 | Theme & Communication | Evaluate the clarity of the subject and its communication. Consider how effectively the narrative, cultural significance, and societal context are conveyed. |
| 6 | Emotion & Viewer Response | Assess how well the work evokes an emotional response, engages the viewer, and creates lasting impressions with personal significance. |
| 7 | Overall Gestalt | Evaluate the overall visual appeal and artistic impact of the image, considering how well the elements combine to create an engaging, meaningful impression. |
| 8 | Comprehensive Evaluation | Provide a comprehensive aesthetics assessment of the image, evaluating its effectiveness in visual impact, theme communication, and artistic depth. |
| – | Overall Aesthetics Score | Overall aesthetics score derived from multi-dimensional evaluation. |

## A.2 CHARATERISTICS OF ARTIMUSE-10K

**WordCloud.** WordCloud of our introduced ArtiMuse-10K dataset is depicted in Fig. 9 . We analyze the textual annotations of ArtiMuse across eight aesthetic attributes and find that the most frequently occurring terms—such as "image," "visual," "composition," "overall," and "elements"—are strongly correlated with image aesthetic quality. This observation suggests that human experts primarily focus on fundamental visual characteristics when assessing artistic merit.

**Score Distributions.** We divide the 10,000 images in the ArtiMuse-10K dataset into a training split (9,000 images) and a test split (1,000 images). The score distributions for both the training and test datasets are shown in Fig. 10. To compare the distribution differences across datasets, we normalize the scores of AVA Murray et al. (2012), PARA Yang et al. (2022), TAD66K He et al. (2022), and FLICKR-AES Ren et al. (2017) to the $[0, 100]$ range and analyze their score distributions, with

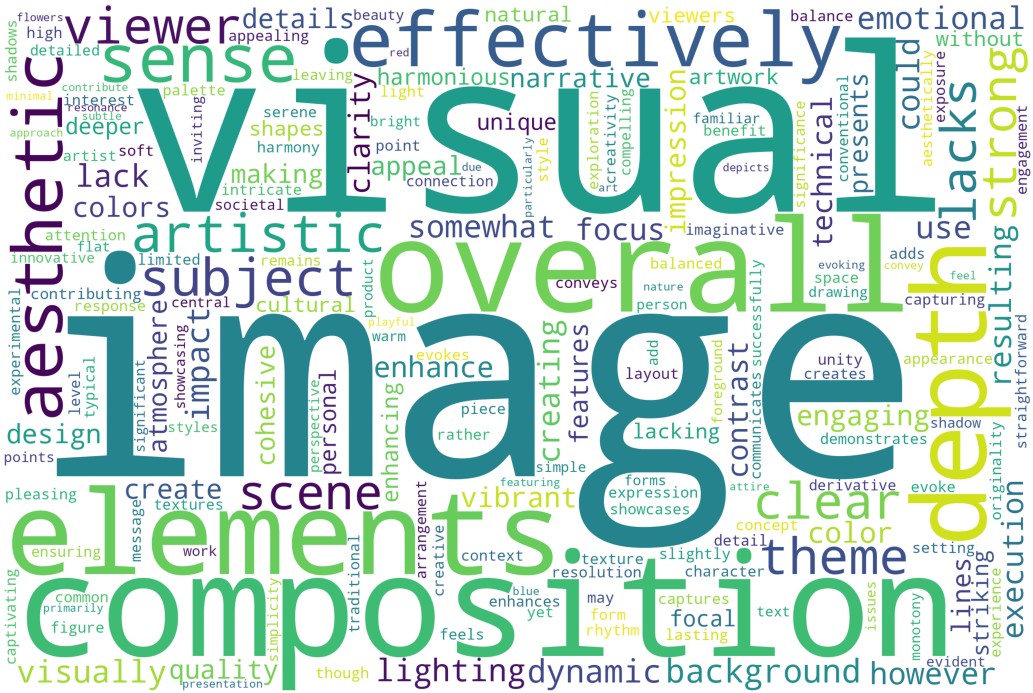

Figure 9: Wordcloud of ArtiMuse-10K dataset. The most frequent words in ArtiMuse-10K dataset are all highly relevant to the aesthetic assessment of images.

results shown in Fig. 11, Fig. 12, Fig. 13, and Fig. 14 respectively. Our analysis reveals that all aesthetic scoring datasets, including ArtiMuse, approximately follow Gaussian distributions. Notably, ArtiMuse demonstrates superior score diversity compared to other datasets.

**Statistical Analysis across Categories.** The images in ArtiMuse-10K are sourced from diverse origins and encompass a total of 5 main categories and 15 subcategories. The detailed distribution of image counts across these categories is presented in Tab. 6.

# B DETAILS OF PUBLIC DATASET COLLECTION & PROCESSING

We select and sample a subset of high-quality aesthetic captions from existing public datasets, with particular emphasis on ensuring both aesthetic caption quality and diversity in the sampling process. The specific sampling statistics are presented in Tab. 7.

## B.1 DATASETS W/ SCORE CAPTION

**AVA Murray et al. (2012), TAD66K He et al. (2022), PARA Yang et al. (2022), FLICKR-AES Ren et al. (2017).** For datasets containing only aesthetic scores without multi-dimensional annotations, we employ the scores as the primary guidance for MLLM to generate comprehensive image evaluations. The following prompt template is adopted:

---
**Aesthetic Score Guidance Prompt**

```
An expert panel award this picture a <score> score out of <range> for
aesthetic quality.  Provide a concise, step-by-step aesthetic analysis
evaluating its strengths and weaknesses in Composition & Design, Visual
Elements & Structure, Technical Execution, Originality & Creativity,
Theme & Communication, Emotion & Viewer Response, Overall Gestalt and
Comprehensive Evaluation.
```

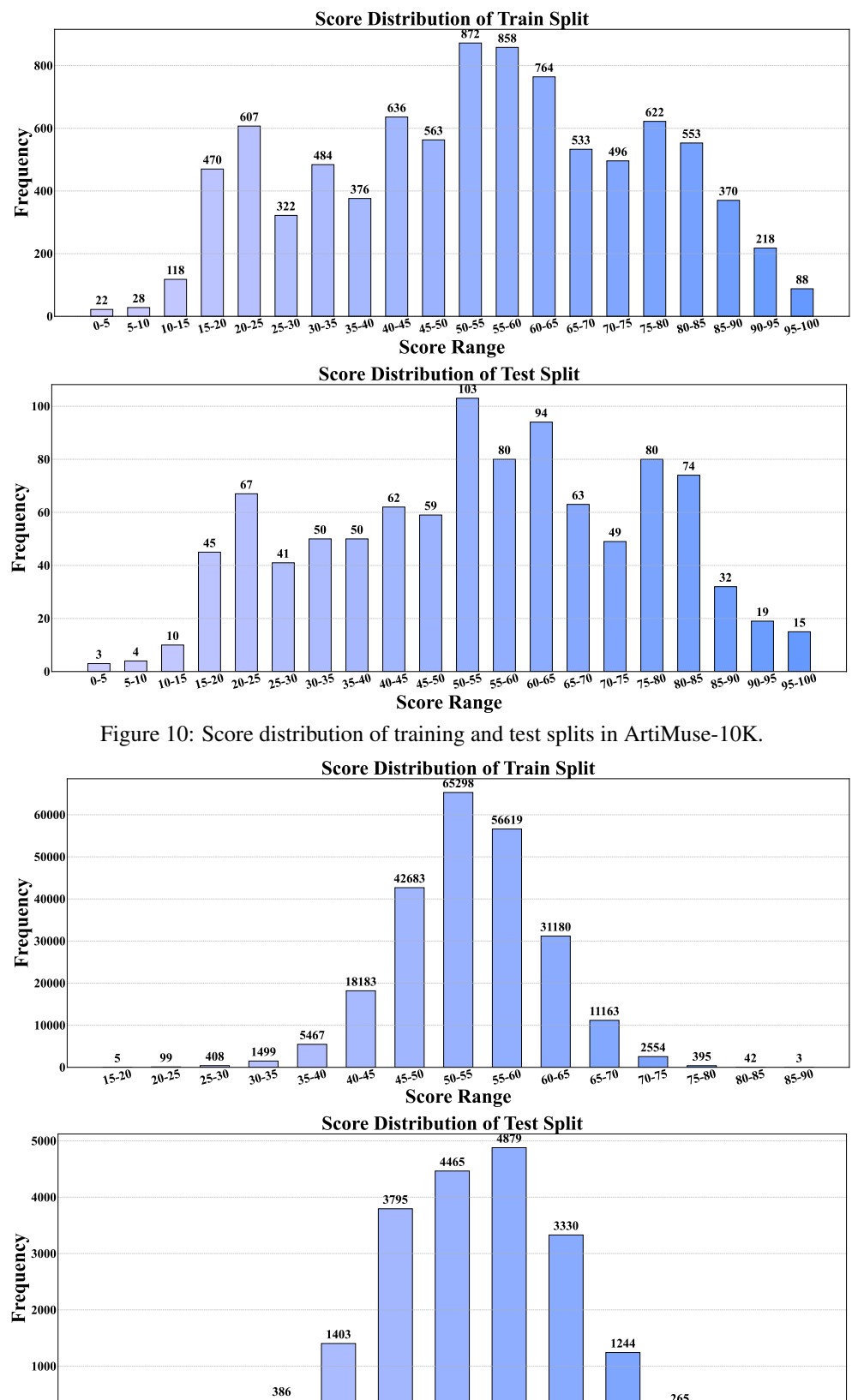

Figure 10: Score distribution of training and test splits in ArtiMuse-10K.

Figure 11: Score distribution of training and test splits in AVA.

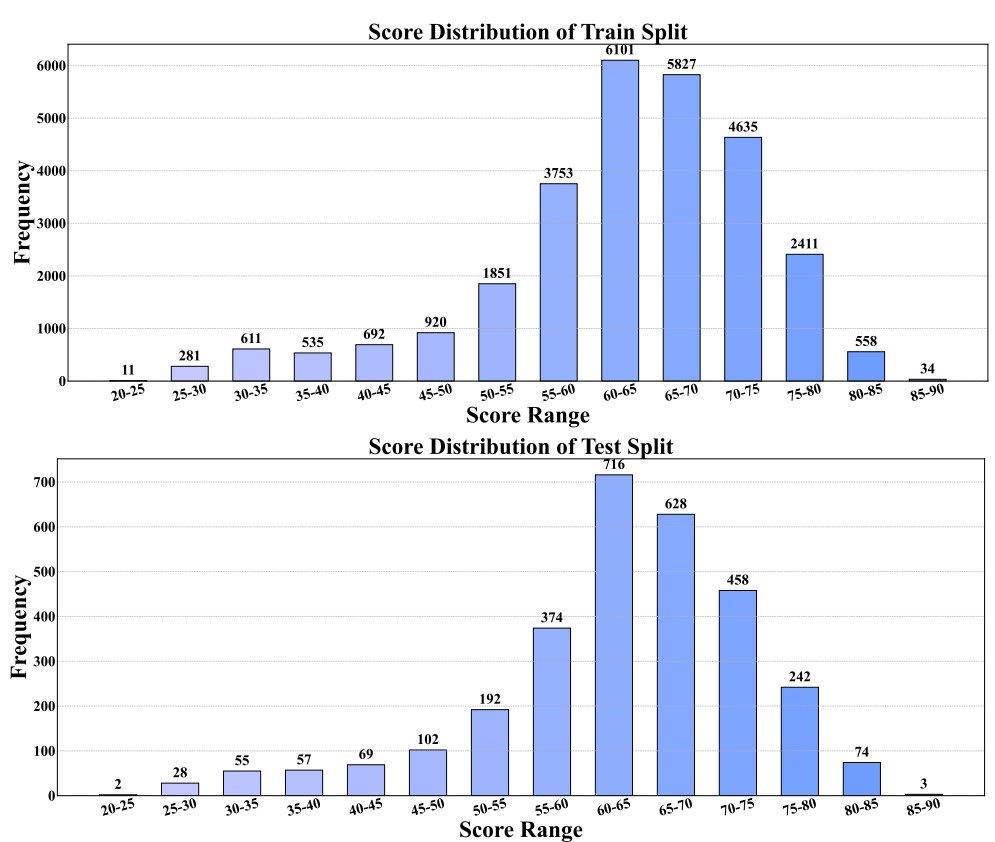

Figure 12: Score distribution of training and test splits in PARA.

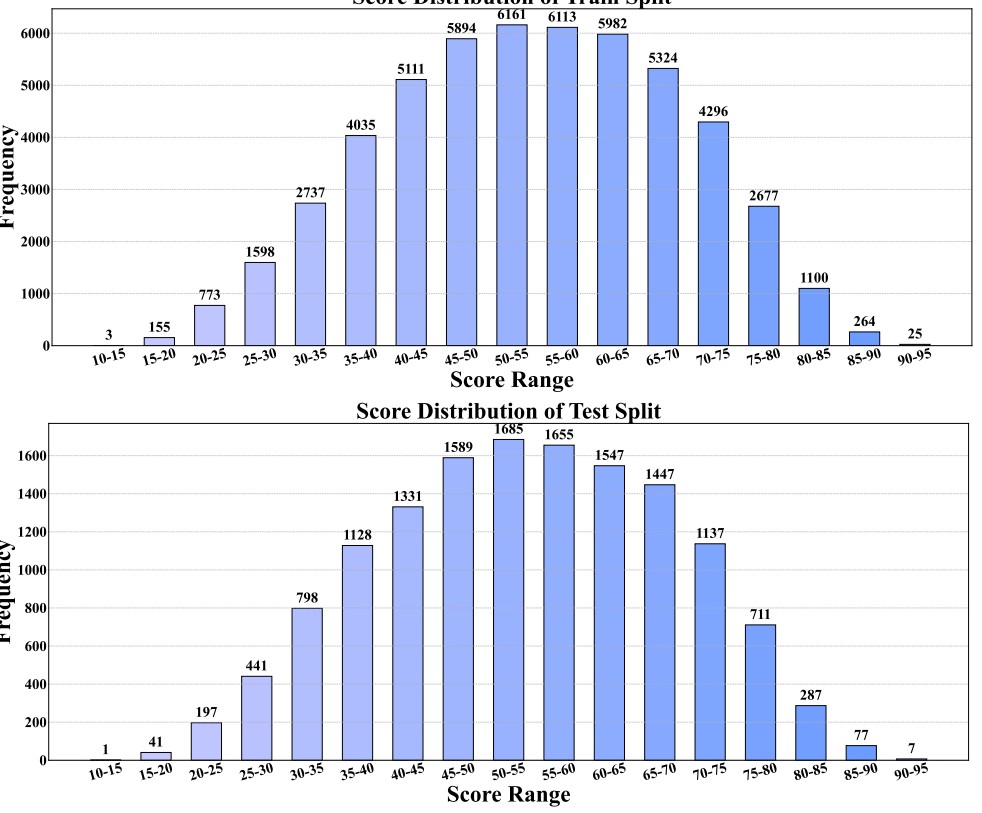

Figure 13: Score distribution of training and test splits in TAD66K.

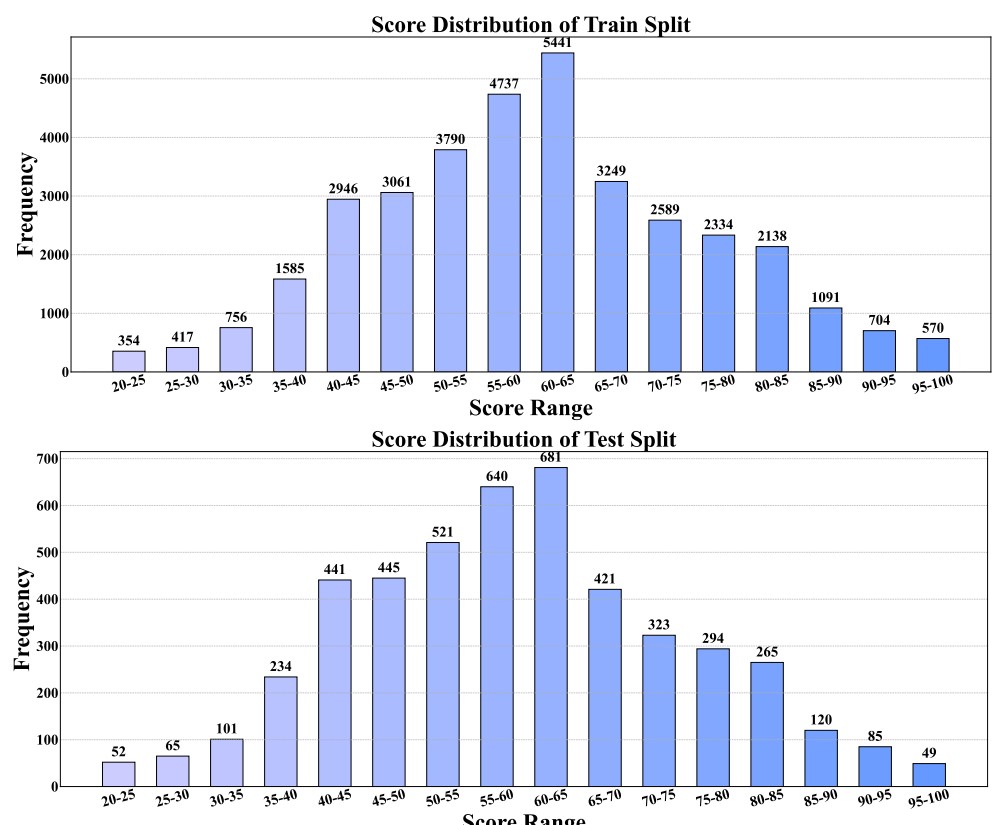

Figure 14: Score distribution of training and test splits in FLICKR-AES.

Table 6: Statistics of ArtiMuse-10K across main categories and subcategories.

| Main Category | Subcategory | Description | # Image |
|---|---|---|---|
| Photography | Daily Photo | Casual photos capturing daily scenes | 3071 |
| | Photographic Art | Photos with artistic processing | 758 |
| | Architecture | Photos of buildings and structures | 119 |
| | Portrait | Portrait photography | 82 |
| | Movie still | Screenshots from films or TV shows | 81 |
| | *Total* | – | **4111** |
| Painting & Calligraphy | Digital Art | Computer-aided digital paintings | 1314 |
| | Children's Painting | Paintings created by children | 699 |
| | Chinese Painting | Chinese ink wash paintings | 511 |
| | General Painting | General paintings with diverse scopes | 485 |
| | Sketch | Pencil/charcoal sketches | 43 |
| | Calligraphy | Artistic handwriting and lettering | 43 |
| | *Total* | – | **3095** |
| AIGC | AIGC | AI-generated content (particularly generative models) | 1453 |
| | *Total* | – | **1453** |
| 3D Design | Product Design | 3D model snapshots for products | 516 |
| | Sculpture | Sculpting artwork snapshots | 307 |
| | *Total* | – | **823** |
| Graphic Design | Graphic Design | Posters/logos/visual designs | 518 |
| | *Total* | – | **518** |
| *Total* | – | – | **10000** |

Table 7: Collection & processing results of public datasets.

| Public Dataset | Dataset Type | Sampled Size |
|---|---|---|
| APDDv2 Jin et al. (2024) | w / partial text caption | 4,898 |
| SPAQ Fang et al. (2020) | w / partial text caption | 1,537 |
| KonIQ-10k Hosu et al. (2020) | w / partial text caption | 1,488 |
| Impressions Kruk et al. (2023) | w / partial text caption | 1,443 |
| AVA Murray et al. (2012) | w / score caption | 235,598 |
| TAD66K He et al. (2022) | w / score caption | 52,248 |
| PARA Yang et al. (2022) | w / score caption | 28,220 |
| FLICKR-AES Ren et al. (2017) | w / score caption | 35,762 |
| *Total* | – | **361,194** |

Here, `<score>` and `<range>` represents the scores and their value ranges, extracted from the dataset with score caption, serve as quantitative indicators to guide the MLLM's image analysis process.

## B.2  DATASETS W/ PARTIAL TEXT CAPTION

**APDDv2 Jin et al. (2024).** The APDDv2 dataset comprises 10,023 images, each annotated with multiple attributes, including: filename, Artistic Categories, Total aesthetic score, Theme and logic, Creativity, Layout and composition, Space and perspective, The sense of order, Light and shadow, Color, Details and texture, The overall, Mood, and Language Comment (the most critical attribute for our study). We filter out samples with excessively short or missing Language Comment entries, retaining 4,898 valid instances. For the filtered data, we design a structured prompt template:

---

**Prompt Template for APDDv2**

```
For the above picture, the artist gave the following evaluation:
<language_comment>.  For other aesthetic attributes:
This image has a artistic category of <artistic_categories>.
The total aesthetic score is <total_aesthetic_score> out of 100.
The score for theme and logic is <theme_and_logic> out of 10.
The score for creativity is <creativity> out of 10.
The score for layout and composition is <layout_and_composition> out of
10.
The score for space and perspective is <space_and_perspective> out of
10.
The score for sense of order is <sense_of_order> out of 10.
The score for light and shadow is <light_and_shadow> out of 10.
The score for color is <color> out of 10.
The score for details and texture is <details_and_texture> out of 10.
The score for overall is <overall> out of 10.
The score for mood is <mood> out of 10.
Please combine the evaluation above with the picture content, then
evaluate the aesthetic quality of this image from the attribute of
<attribute>.  <description>.  Limit the assessment to one paragraph
(<=100 words), avoiding markdown formatting.  Answer in English.  Do
not repeat contents in artist's evaluation (like scores).
```

---

which incorporates key information such as the overall comment, category labels, and subcategory scores to ensure comprehensive utilization of the available annotations. Here, words enclosed in angle brackets (<>) denote referenced phrases or statements. For instance, `<language_comment>`, `<artistic_categories>`, `<total_aesthetic_score>`, ..., and `<mood>` refer to the corresponding captions in the dataset, while `<attribute>` and `<description>` represent the specific attribute and its description listed in Tab. A.1.

**SPAQ Fang et al. (2020).** The original dataset contains various image attributes, including EXIF tags, mean opinion scores (MOS), image attribute scores, and scene category labels. The SPAQ

dataset comprises 11,125 images, which we filter according to two key criteria: (1) 80% of the filtered subset must have either MOS (Mean Opinion Score) or the average of four quality metrics (brightness, colorfulness, contrast, and sharpness) falling within the extreme ranges of $[0, 25]$ or $[75, 100]$, ensuring sufficient representation of both low and high aesthetic quality samples; (2) all selected images must contain valid entries for the "categories" attribute. From these, we select attributes relevant to visual aesthetics—specifically, MOS ratings and a subset of aesthetic-related attribute scores—and designed the following prompt template:

---

**Prompt Template for SPAQ**

```
The score for overall quality is <mos> out of 100, with a high degree
(if <mos> > 75) / low degree (if <mos> < 25) of aesthetic appeal.
The score for brightness is <brightness> out of 100.
The score for colorfulness is <colorfulness> out of 100.
The score for contrast is <contrast> out of 100.
The score for sharpness is <sharpness> out of 100.
The image content belongs to the following categories:  <categories>.
Please combine the evaluation above with the picture content, then
evaluate the aesthetic quality of this image from the attribute of
<attribute>.  <description>.  Limit the assessment to one paragraph
(<=100 words), avoiding markdown formatting.  Answer in English.  Do
not repeat contents in artist's evaluation (like scores).
```

---

Here, The classification into high-degree and low-degree categories is governed by the MOS threshold: instances with MOS > 75 are designated as high-degree, while those with MOS < 25 are categorized as low-degree. The placeholders `<mos>`, `<brightness>`, `<colorfulness>`, ..., and `<categories>` correspond to the respective captions from the SPAQ dataset, while `<attribute>` and `<description>` refer to the specific aesthetic attributes and their detailed descriptions as presented in Table A.1.

**KonIQ-10K Hosu et al. (2020).** The KonIQ-10K dataset comprises 10,000 images, from which we select the following aesthetic-relevant attributes for filtering: MOSz, brightness, contrast, colorfulness, sharpness, and quality_factor. Our filtering criteria requires that 80% of the selected images must have MOSz scores falling within either the $[0, 25]$ or $[75, 100]$ ranges, ensuring balanced representation of both low and high aesthetic quality samples. Through this process, we obtain 1,488 filtered images, which are then annotated by the MLLM using the following prompt template:

---

**Prompt Template for KonIQ-10K**

```
The score for overall quality is <MOSz> out of 100, with a high degree
(if <MOSz> > 75) / low degree (if <MOSz> < 25) of aesthetic appeal.
The score for brightness is <brightness> out of 1.
The score for contrast is <contrast> out of 1.
The score for colorfulness is <colorfulness> out of 1.
The score for sharpness is <sharpness> out of 100.
Please combine the evaluation above with the picture content, then
evaluate the aesthetic quality of this image from the attribute of
<attribute>.  <description>.  Limit the assessment to one paragraph
(<=100 words), avoiding markdown formatting.  Answer in English.  Do
not repeat contents in artist's evaluation (like scores).
```

---

Here, The classification into high-degree and low-degree categories is governed by the MOSz threshold: instances with MOSz > 75 are designated as high-degree, while those with MOSz < 25 are categorized as low-degree. The placeholders `<MOSz>`, `<brightness>`, `<contrast>`, ..., and `<sharpness>` correspond to the respective captions from the KonIQ-10K dataset, while `<attribute>` and `<description>` refer to the specific aesthetic attributes and their detailed descriptions as presented in Table A.1.

**Impressions Kruk et al. (2023).** The original dataset contains over 1,400 images, each accompanied by multiple annotations (including image descriptions, impressions, and aesthetic evaluations) from different annotators, resulting in more than 4,800 data entries in total. Along with these annotations, Impressions also collects detailed annotator metadata such as educational background and aesthetic experience. To ensure annotation quality, we apply the following filtering criterion: for

each image, we retain only the evaluation from the most aesthetically experienced annotator. This filtering process yields a refined dataset of 1,443 high-quality annotations, which are then annotated by the MLLM using the following prompt template:

---
**Prompt Template for Impressions**

```
This image's caption is: <caption>.
What is happening in the image: .
The emotions/thoughts/beliefs that the photograph may inspire:
<image_impression>.
The aesthetic elements that elicited the expressed impression:
<image_aesthetic_eval>.
Please combine the evaluation above with the picture content, then
evaluate the aesthetic quality of this image from the attribute of
<attribute>.  <description>.  Limit the assessment to one paragraph
(<=100 words), avoiding markdown formatting.  Answer in English.  Do
not repeat contents in artist's evaluation (like scores).
```
---

Here, the placeholders *<caption>*, **, *<image_impression>*, and *<image_aesthetic_eval>* correspond to the respective captions from the Impressions dataset, while *<attribute>* and *<description>* refer to the specific aesthetic attributes and their detailed descriptions as presented in Table A.1.

## C    DETAILS OF TOKEN AS SCORE STRATEGY

We conducted a comprehensive comparison of various score prediction strategies, and the experimental results are presented in Tab. 10. Across all experiments, the prediction score methodology was the sole differentiating factor, while the training data, training configurations, and model architecture remained consistent. To ensure robust and reliable experimental conclusions, we conduct comprehensive evaluations on both AVA (the largest image aesthetics scoring dataset) Murray et al. (2012) and ArtiMuse-10K (ours).

### C.1    LEVEL AS SCORE

Following Q-Align Wu et al. (2024b), we predict scores by predicting five distinct discrete levels. Specifically, during training, we convert the continuous scores in the dataset into corresponding levels based on a predefined mapping and train the model to predict these discrete levels. This mapping scheme involves uniformly dividing the range between the maximum score (M) and the minimum score (m) into five distinct intervals, with scores within each interval being assigned to a corresponding discrete level:

$$L(s) = l_i \text{ if } m + \frac{i-1}{5} \times (M-m) < s \le m + \frac{i}{5} \times (M-m) \tag{2}$$

where

$$\{l_i\}_{i=1}^5 = \{\text{bad, poor, fair, good, excellent}\} \tag{3}$$

which are the standard text rating levels as defined by ITU BT (2002). During inference, the final score prediction was derived by computing a weighted sum of the predicted probability distribution across these five levels.

**Discussions.** The comparison between Exp. (a) and (i) in Tab. 10, along with other experimental groups, demonstrates that the Level As Score approach exhibits a significant performance degradation compared to the Token As Score. This decline can be attributed to the overly coarse-grained level partitioning scheme, which fails to achieve fine-grained score mapping. Furthermore, the adopted vocabulary lacks proper alignment with the LLM's lexical table design, collectively contributing to the suboptimal outcomes.

### C.2    TOKEN AS SCORE W/ EXPANDING TOKENS

We provide a detailed exposition of the *Token As Score* strategy, as referenced in the Sec. 4.3 of the main paper. In this investigation, we explore the expansion of the LLM vocabulary

by incorporating additional tokens specifically for aesthetics score prediction. For instance, in the "Expanding 25 Tokens" configuration, we augment the vocabulary with the following tokens: [AES_SCORE_TOKEN_0], [AES_SCORE_TOKEN_1], [AES_SCORE_TOKEN_2], ..., [AES_SCORE_TOKEN_25]. These tokens correspond to predicted scores of 0, 4, 8, ..., 100, respectively. The model is trained to predict these specialized tokens, and during inference, the final aesthetic score is derived by computing a weighted sum based on the predicted probability distribution over these tokens.

**Discussions.** A comparison of experiments (b)-(f) on AVA reveals that the performance of the Token As Score strategy initially improves and then declines as the number of introduced tokens increases, peaking at 100 tokens. This trend occurs because an insufficient number of tokens fails to establish an accurate token-score mapping, while an excessive number exceeds the available data or model capacity, leading to underfitting. Experimental results on ArtiMuse-10K demonstrate that the Token As Score approach with expanding tokens performs poorly, suggesting this method fails to converge properly when either the dataset is inherently challenging or insufficient in size.

C.3 TOKEN AS SCORE W/ EXISTING TOKENS

We futher explore the selection of a subset of the LLM's existing displayable tokens for aesthetics score prediction. Our selection criteria prioritize brevity, inherent order, ease of convergence during training, and minimal ambiguity with numerical scores. As illustrated in Tab. 10, our specific configurations in experiments are as follows:

**Existing 25 Tokens.** We select the tokens a, b, c, ... , y, which are sequentially mapped to scores ranging from 0 to 100 with an interval of 4 (i.e., 0, 4, 8, ..., 100).

**Existing 50 Tokens.** We select the tokens a, b, c, ... , y, A, B, C, ... , Y, which are sequentially mapped to scores ranging from 0 to 100 with an interval of 2 (i.e., 0, 2, 4, ..., 100).

**Existing 100 Tokens (non-ordered).** We select the first 100 character tokens starting from 0 within the vocabulary of the Qwen2.5-7B LLM, as detailed in Tab. 8. These tokens are sequentially mapped to scores from 0 to 100.

Table 8: Token-score mapping table for existing 100 tokens (non-ordered).

| Token ID | 15 | 16 | 17 | 18 | 19 | 20 | 21 | 22 | 23 | 24 | 25 | 26 | 27 | 28 | 29 | 30 | 31 | 32 | 33 | 34 |
|---|---|---|---|---|---|---|---|---|---|---|---|---|---|---|---|---|---|---|---|---|
| Token | 0 | 1 | 2 | 3 | 4 | 5 | 6 | 7 | 8 | 9 | : | ; | < | = | > | ? | @ | A | B | C |
| Score | 0 | 1 | 2 | 3 | 4 | 5 | 6 | 7 | 8 | 9 | 10 | 11 | 12 | 13 | 14 | 15 | 16 | 17 | 18 | 19 |
| **Token ID** | 35 | 36 | 37 | 38 | 39 | 40 | 41 | 42 | 43 | 44 | 45 | 46 | 47 | 48 | 49 | 50 | 51 | 52 | 53 | 54 |
| Token | D | E | F | G | H | I | J | K | L | M | N | O | P | Q | R | S | T | U | V | W |
| Score | 20 | 21 | 22 | 23 | 24 | 25 | 26 | 27 | 28 | 29 | 30 | 31 | 32 | 33 | 34 | 35 | 36 | 37 | 38 | 39 |
| **Token ID** | 55 | 56 | 57 | 58 | 59 | 60 | 61 | 62 | 63 | 64 | 65 | 66 | 67 | 68 | 69 | 70 | 71 | 72 | 73 | 74 |
| Token | X | Y | Z | [ | \ | ] | ^ | _ | ` | a | b | c | d | e | f | g | h | i | j | k |
| Score | 40 | 41 | 42 | 43 | 44 | 45 | 46 | 47 | 48 | 49 | 50 | 51 | 52 | 53 | 54 | 55 | 56 | 57 | 58 | 59 |
| **Token ID** | 75 | 76 | 77 | 78 | 79 | 80 | 81 | 82 | 83 | 84 | 85 | 86 | 87 | 88 | 89 | 90 | 91 | 92 | 93 | 94 |
| Token | l | m | n | o | p | q | r | s | t | u | v | w | x | y | z | { | ‖ | } | ~ | ¡ |
| Score | 60 | 61 | 62 | 63 | 64 | 65 | 66 | 67 | 68 | 69 | 70 | 71 | 72 | 73 | 74 | 75 | 76 | 77 | 78 | 79 |

| Token ID | 95 | 96 | 97 | 98 | 99 | 100 | 101 | 102 | 103 | 104 | 105 | 106 | 107 | 108 | 109 | 110 | 111 | 112 | 113 | 114 | 115 |
|---|---|---|---|---|---|---|---|---|---|---|---|---|---|---|---|---|---|---|---|---|---|
| Token | ¢ | £ | ¤ | ¥ | ¦ | § | ¨ | © | ª | « | ¬ | ® | ¯ | ° | ± | ² | ³ | ´ | µ | ¶ | · |
| Score | 80 | 81 | 82 | 83 | 84 | 85 | 86 | 87 | 88 | 89 | 90 | 91 | 92 | 93 | 94 | 95 | 96 | 97 | 98 | 99 | 100 |

**Existing 100 Tokens (ordered).** This represents the final approach adopted in ArtiMuse. We construct 100 tokens by concatenating lowercase letters, ensuring these tokens are ordered within the vocabulary of the Qwen2.5-7B LLM, as presented in Tab. 9. These tokens are sequentially mapped to scores from 0 to 100.

**Discussions.** The comparisons in (b)-(g), (c)-(h), and (d)-(j) demonstrate that when using the same number of tokens for prediction in the Token As Score, tokens from the existing vocabulary consistently yield better performance. This occurs because newly introduced tokens lack corresponding prior knowledge from the model's pretraining phase and do not possess inherent ordinal relationships with scores, making them less effective than tokens in the LLM vocabulary that carry clear semantic information and sequential relationships.

Table 9: Token-score mapping table for existing 100 tokens (ordered), which is used in ArtiMuse.

| Token | aa | ab | ac | ad | ae | af | ag | ah | ai | aj | ak | al | am | an | ao | ap | aq | ar | as | at |
|-------|----|----|----|----|----|----|----|----|----|----|----|----|----|----|----|----|----|----|----|----|
| Score | 0 | 1 | 2 | 3 | 4 | 5 | 6 | 7 | 8 | 9 | 10 | 11 | 12 | 13 | 14 | 15 | 16 | 17 | 18 | 19 |
| Token | au | av | aw | ax | ay | az | ca | cb | cc | cd | ce | cf | cg | ch | ci | cj | ck | cl | cm | cn |
| Score | 20 | 21 | 22 | 23 | 24 | 25 | 26 | 27 | 28 | 29 | 30 | 31 | 32 | 33 | 34 | 35 | 36 | 37 | 38 | 39 |
| Token | co | cp | cq | cr | cs | ct | cu | cv | cw | cx | cy | da | db | dc | dd | de | df | dg | dh | di |
| Score | 40 | 41 | 42 | 43 | 44 | 45 | 46 | 47 | 48 | 49 | 50 | 51 | 52 | 53 | 54 | 55 | 56 | 57 | 58 | 59 |
| Token | dj | dk | dl | dm | dn | do | dp | dq | dr | ds | dt | du | dv | dw | dx | dy | ea | eb | ec | ed |
| Score | 60 | 61 | 62 | 63 | 64 | 65 | 66 | 67 | 68 | 69 | 70 | 71 | 72 | 73 | 74 | 75 | 76 | 77 | 78 | 79 |
| Token | ee | ef | eg | eh | ei | ej | ek | el | em | en | eo | ep | eq | er | es | et | eu | ev | ew | ey |
| Score | 80 | 81 | 82 | 83 | 84 | 85 | 86 | 87 | 88 | 89 | 90 | 91 | 92 | 93 | 94 | 95 | 96 | 97 | 98 | 100 |

Furthermore, experiments (g), (h), and (j) reveal that when using existing tokens for Token As Score, model performance improves significantly as the number of tokens increases. Due to the limited number of displayable characters in the Qwen2.5-7B LLM vocabulary, we are currently unable to further increase this quantity, which will be explored in future work. Additionally, comparing (i) and (j) shows that the choice of tokens also affects performance—the token mapping scheme in (j), which has more explicit semantic and ordinal relationships, leads to better results.

Table 10: Explorations on score prediction strategies. To ensure experimental validity, we conduct our experiments both on the AVA dataset and AriMuse-10K dataset. (j) represents the setting of Token As Score strategy in ArtiMuse. Beyond the convergence issues observed with the expanding strategy on ArtiMuse-10K, the 100-token configuration demonstrates peak performance across various token quantities.

| Exp. | Score Prediction | AVA Murray et al. (2012) | | ArtiMuse-10K | |
|------|------------------|------|------|------|------|
| | | SRCC | PLCC | SRCC | PLCC |
| (a) | 5 Levels | 0.820 | 0.818 | 0.571 | 0.551 |
| (b) | Expanding 25 Tokens | 0.803 | 0.665 | 0.045 | 0.055 |
| (c) | Expanding 50 Tokens | 0.822 | 0.821 | 0.018 | 0.027 |
| (d) | Expanding 100 Tokens | 0.824 | 0.822 | 0.029 | 0.027 |
| (e) | Expanding 250 Tokens | 0.823 | 0.821 | -0.012 | 0.002 |
| (f) | Expanding 500 Tokens | 0.821 | 0.819 | 0.006 | 0.012 |
| (g) | Existing 25 Tokens | 0.823 | 0.822 | 0.006 | 0.010 |
| (h) | Existing 50 Tokens | 0.825 | 0.824 | 0.612 | 0.623 |
| (i) | Existing 100 Tokens (non-ordered) | 0.826 | 0.825 | 0.582 | 0.541 |
| (j) | Existing 100 Tokens (ordered) | **0.827** | **0.826** | **0.614** | **0.627** |

## D  IMPLEMENTATION DETAILS

### D.1  TRAINING DETAILS

**Hyperparameters.** We employ the InternVL-3-8B Zhu et al. (2025) model as our base model and adopt its default hyperparameters for the aesthetic assessment task through two training stages: Text Pretrain and Score Finetune. The pre-trained models and specific hyperparameter configurations are detailed in Table 11, with modifications carefully designed to address the unique requirements of visual aesthetic evaluation.

**Resolution Strategy.** The original InternVL-3 model employs a dynamic high-resolution strategy Zhu et al. (2025) to handle images of varying resolutions and attribute ratios. This approach involves three key steps: closest attribute ratio matching, image resizing and splitting, and optional thumbnail generation. Given an input image with dimensions $W \times H$, the aspect ratio $r = W/H$ is computed. The algorithm selects a target aspect ratio $r_{best}$ from a predefined set $\mathcal{R}$, which minimizes distortion while constraining the number of tiles $n_{tiles}$ within a range $[n_{min}, n_{max}]$. The image is resized to dimensions $S \times i_{best} \times S \times j_{best}$ (where $S = 448$) and split into $n_{tiles} = i_{best} \times j_{best}$ tiles of size $S \times S$. If $n_{tiles} > 1$, a thumbnail of size $S \times S$ is appended to preserve a global view.

Table 11: Pre-trained models and hyperparameters used for ArtiMuse, including text pretraining and score finetuning.

| Pre-trained models / Hyperparameters | Text Pretrain | Score Finetune |
|---|---|---|
| Vison Encoder | InternViT-300M-448px-V2.5 | InternViT-300M-448px-V2.5 |
| Large Language Model | Qwen2.5-7B | Qwen2.5-7B |
| Large Language Model LoRA Rank | 16 | 128 |
| Image Resolution | $448 \times 448$ | $448 \times 448$ |
| Max Sequence Length | 8192 | 8192 |
| Batch Size | 128 | 128 |
| Warmup Epochs | 0.03 | 0.03 |
| Gradient Accuracy | 1 | 1 |
| Numerical Precision | Float16 | Float16 |
| LR Schedule | Cosine decay | Cosine decay |
| LR Max | 4e-5 | 2e-5 |
| Weight Decay | 0.05 | 0 |
| Epoch | 1 | 2 |

However, in ArtiMuse, we adopt a fixed-resolution strategy instead of the dynamic approach. Aesthetic evaluation relies heavily on holistic image features, such as composition, color harmony, and spatial relationships, which can be disrupted by splitting an image into localized tiles. The dynamic strategy's tile-based processing risks fragmenting these global characteristics, thereby degrading performance in tasks requiring an integrated understanding of visual aesthetics. By resizing all images to a uniform resolution without tiling, we preserve the structural and semantic coherence of the entire image. This adjustment ensures that the model captures aesthetic qualities through a consistent, undistorted representation of the input, aligning better with the requirements of fine-grained aesthetic analysis. Our experiments demonstrate that employing the fixed-resolution strategy yields approximately 0.3 improvements in both SRCC and PLCC metrics for aesthetic scoring tasks compared to the dynamic high-resolution strategy, while simultaneously more than doubling training and inference efficiency.

### D.2 INFERENCE DETAILS FOR AESTHETICS SCORING

We present the implementation details for various models in the aesthetic scoring task. Note that certain models—including TANet He et al. (2022), AesMamba Gao et al. (2024), UNIAA-LLaVA Zhou et al. (2024), and Next Token Is Enough Li et al. (2025)—are excluded from this discussion due to testing constraints.

**Models w/ Scoring Ability.** For models capable of generating aesthetic scores (Q-Instruct Wu et al. (2023), PEAS Yun & Choo (2024a), Q-Align Wu et al. (2024b)), we directly utilize their scoring outputs. In cases where a model provides only general assessments (MUSIQ Ke et al. (2021)), we adopt its general score as the final evaluation result.

**Models w/o Scoring Ability.** For models lacking inherent scoring capabilities (VILA Lin et al. (2023), mPLUG-Owl2 Ye et al. (2023), ShareGPT-4V Chen et al. (2024), Qwen-2.5-VL-7B Bai et al. (2025), InternVL3-8B Zhu et al. (2025)), we employ carefully designed prompts to elicit numerical evaluations. The prompt structure is as follows:

---
**Prompts for Models without Scoring Ability**

```
Please rate the aesthetic quality of this image and provide a score
between 0 and 100, where 0 represents the lowest quality and 100
represents the highest.  Your response should contain only an integer
value.
```
---

This prompt guides the model to output an integer score from 0 to 100, aligning with ArtiMuse's scoring format. We use these prompted scores for comparative analysis, ensuring consistency across all evaluated models.

### D.3 Inference Details for Textual Analysis

When evaluating the model's textual analysis capability, we design specialized prompts for comparative models by incorporating relevant aesthetic background knowledge to ensure fairness. Specifically, for ArtiMuse, we employ the following prompt format during testing:

> **Prompts for ArtiMuse**
>
> ```
> Please evaluate the aesthetic quality of this image from the attribute
> of <attribute>.
> ```

where `<attribute>` represents the specific attribute listed in Tab. A.1. For other models, we augment their inputs with corresponding attribute descriptions to maintain parity in contextual understanding:

> **Prompts for Other Models**
>
> ```
> Background Knowledge: <attribute>: <description>. Please evaluate
> the aesthetic quality of this image from the attribute of <attribute>.
> No more than 100 words.
> ```

where `<attribute>` and `<description>` represent the specific attribute and its description listed in Tab. A.1. Additional textual evaluation results and analysis are presented in Section E.5.

### D.4 Comparison Details

**Judging by MLLM.** We provide a detailed explanation of the methodology employed in Sec. 5.2 of the main paper for using MLLMs to select among different models' structural aesthetic analysis results. As illustrated in Fig. 15, we first determine the input image and the corresponding aesthetic attributes, then guide the MLLM to generate textual evaluations using the following prompt template:

> **Prompts for Generating Textual Evaluation**
>
> ```
> You are an aesthetic evaluation expert. Please evaluate the aesthetic
> quality of this image from the attribute of <attribute>. No more than
> 100 words.
> ```

where `<attribute>` corresponds to the specific aesthetic attributes listed in Tab. A.1. For human experts, we also provide the attribute and invite them to provide textual evaluations. The image, attribute, expert evaluations, and the outputs from different models are then fed into a judgment MLLM (specifically, Gemini-2.0-flash) for assessment. We guide this MLLM to evaluate and select the highest-quality responses among the model outputs using a single-choice question format prompt (Taking 4 models as an example):

> **MLLM-as-Judge Prompts**
>
> ```
> You are an expert aesthetic evaluation judge. Your task is to evaluate
> the aesthetic analysis quality of each model's response, based on
> its alignment with the given human expert critique. There are four
> model-generated responses: model1, model2, model3, and model4.
> Assess them independently for clarity, accuracy, insightfulness, and
> relevance, and identify the single best response overall. Output
> only the identifier of the best model (i.e., one of: model1, model2,
> model3, model4) -- do not include any extra text, explanation, symbols,
> or formatting.
> ```

which minimizes hallucinations, provides sufficient information for decision-making, and ensures consistent evaluation criteria across all model responses, thereby yielding relatively accurate and stable selection outcomes. The results is presented in Tab. 2 of the main paper.

**Judging by Human.** For the user study, we randomly select 20 images from the ArtiMuse-10K test set, ensuring coverage across different categories and varying aesthetic qualities. Each image

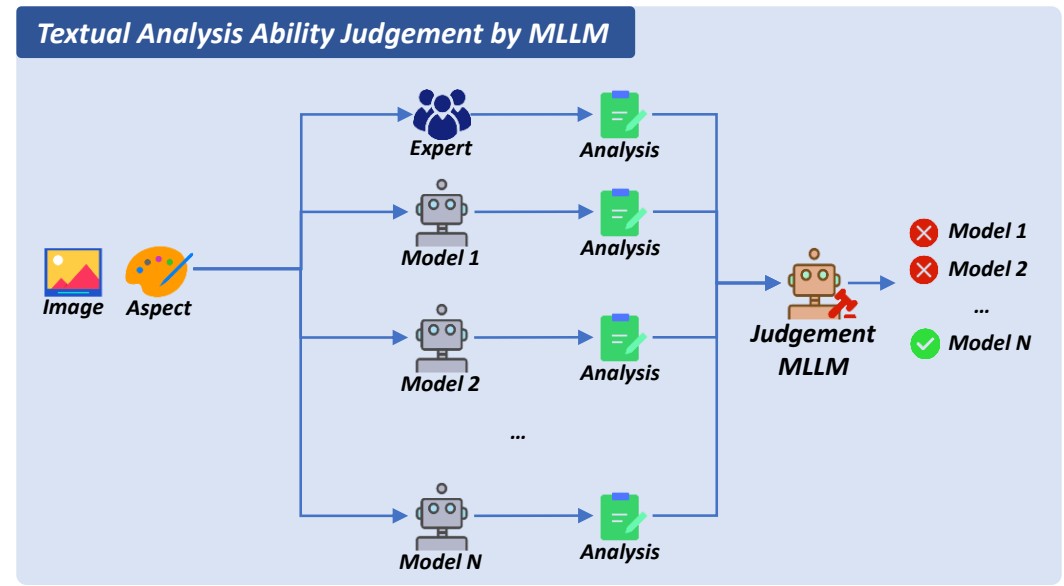

Figure 15: Pipeline of the structural aesthetic analysis ability judgment by MLLM.

is evaluated by different models across 8 aesthetic attributes, with their outputs recorded. We compile these results into 20 multiple-choice questions, where each question corresponds to one image and the model-generated evaluations for a specific attribute, supplemented by a detailed description of that attribute for context. We recruit 20 volunteers, including both individuals without formal training and those with extensive aesthetic evaluation experience, to participate in the study. Their selections are collected, and the preference rates for each model are computed. The results are presented in Tab. 2 of the main paper.

# E   MORE RESULTS

## E.1   COMPARISON WITH SOTA OPEN-SOURCE & CLOSED-SOURCE MLLMS

We benchmark ArtiMuse against state-of-the-art multimodal large language models (MLLMs), including both open-source (Qwen-2.5-VL-72B-instruct Bai et al. (2025) and InternVL3-78B Zhu et al. (2025)) and closed-source models (GPT-4o Achiam et al. (2023) and Gemini-2.0-Flash Team et al. (2023)). As shown in Tab. 12, closed-source models generally outperform open-source models. Notably, ArtiMuse achieves significantly higher performance in aesthetics scoring than these leading MLLMs despite having only 8B parameters, demonstrating its exceptional capability in image aesthetic assessment.

Table 12: More comparison on aesthetics scoring. The best and second-best performances are highlighted in red and blue, respectively. ArtiMuse demonstrates superior performance when compared to various state-of-the-art open-source & closed-source MLLMs.

| Model | AVA Murray et al. (2012) | | PARA Yang et al. (2022) | | TAD66K He et al. (2022) | | FLICKR-AES Ren et al. (2017) | | ArtiMuse-10K | |
|---|---|---|---|---|---|---|---|---|---|---|
| | SRCC | PLCC | SRCC | PLCC | SRCC | PLCC | SRCC | PLCC | SRCC | PLCC |
| *Comparison with SOTA Open-Source & Closed-Source MLLMs* | | | | | | | | | | |
| Qwen-2.5-VL-72B-instruct Bai et al. (2025) | 0.408 | 0.387 | 0.727 | 0.763 | 0.232 | 0.235 | 0.626 | 0.589 | 0.233 | 0.197 |
| InternVL3-78B Zhu et al. (2025) | 0.385 | 0.344 | 0.666 | 0.694 | 0.221 | 0.220 | 0.518 | 0.433 | 0.223 | 0.206 |
| GPT-4o Achiam et al. (2023) | 0.509 | 0.485 | 0.697 | 0.744 | 0.278 | 0.282 | 0.605 | 0.597 | 0.333 | 0.276 |
| Gemini-2.0-flash Team et al. (2023) | 0.474 | 0.457 | 0.703 | 0.704 | 0.319 | 0.323 | 0.658 | 0.651 | 0.286 | 0.265 |
| **ArtiMuse (Ours)** | **0.827** | **0.826** | **0.936** | **0.958** | **0.510** | **0.543** | **0.814** | **0.837** | **0.614** | **0.627** |

## E.2   FURTHER COMPARISON OF GENERALIZATION ABILITY

We further experimentally validate ArtiMuse's generalization ability through comprehensive cross-dataset evaluations. As shown in Tab. 13, we train both the state-of-the-art open-source IAA model

Q-Align Wu et al. (2024b) and ArtiMuse on AVA Murray et al. (2012), PARA Yang et al. (2022), TAD66K He et al. (2022), FLICKR-AES Ren et al. (2017), and ArtiMuse-10K, then evaluate them across all five datasets. The results demonstrate that ArtiMuse consistently outperforms Q-Align on unseen datasets in most cases, confirming its superior generalization capability.

Table 13: Further comparison of generalization ability. The best performances are highlighted in red. * Results are trained only on single dataset to compare the generalization ability. ArtiMuse demonstrates strong generalization capabilities when compared to state-of-the-art IAA models.

| Model | AVA Murray et al. (2012) | | PARA Yang et al. (2022) | | TAD66K He et al. (2022) | | FLICKR-AES Ren et al. (2017) | | ArtiMuse-10K | |
|---|---|---|---|---|---|---|---|---|---|---|
| | SRCC | PLCC | SRCC | PLCC | SRCC | PLCC | SRCC | PLCC | SRCC | PLCC |
| *Further Comparison of Generalization Ability* | | | | | | | | | | |
| Q-Align (AVA) * | 0.822 | 0.817 | 0.694 | 0.711 | 0.417 | 0.445 | 0.643 | 0.664 | 0.337 | 0.320 |
| **ArtiMuse (AVA) *** | 0.827 | 0.826 | 0.697 | 0.725 | 0.419 | 0.451 | 0.647 | 0.676 | 0.395 | 0.376 |
| Q-Align (PARA) * | 0.492 | 0.456 | 0.913 | 0.888 | 0.300 | 0.281 | 0.913 | 0.888 | 0.158 | 0.115 |
| **ArtiMuse (PARA) *** | 0.493 | 0.510 | 0.936 | 0.958 | 0.301 | 0.311 | 0.936 | 0.958 | 0.229 | 0.188 |
| Q-Align (TAD66K) * | 0.695 | 0.699 | 0.688 | 0.667 | 0.501 | 0.531 | 0.688 | 0.667 | 0.317 | 0.304 |
| **ArtiMuse (TAD66K) *** | 0.671 | 0.676 | 0.719 | 0.677 | 0.510 | 0.543 | 0.719 | 0.677 | 0.397 | 0.369 |
| Q-Align (FLICKR-AES) * | 0.609 | 0.611 | 0.836 | 0.839 | 0.366 | 0.376 | 0.798 | 0.818 | 0.215 | 0.208 |
| **ArtiMuse (FLICKR-AES) *** | 0.581 | 0.594 | 0.854 | 0.874 | 0.379 | 0.397 | 0.814 | 0.837 | 0.294 | 0.285 |
| Q-Align (ArtiMuse-10K) * | 0.398 | 0.386 | 0.346 | 0.395 | 0.194 | 0.197 | 0.137 | 0.123 | 0.551 | 0.573 |
| **ArtiMuse (ArtiMuse-10K) *** | 0.397 | 0.385 | 0.446 | 0.461 | 0.230 | 0.232 | 0.349 | 0.334 | 0.614 | 0.627 |

### E.3    IMAGE EXAMPLES IN ARTIMUSE-10K

As illustrated in Fig. 16, Fig. 17 and Fig. 18, the ArtiMuse-10K dataset includes a diverse collection of images, meticulously organized across all specified subcategories. The dataset encompasses a wide range of aesthetic qualities and sources, ensuring rich variability and broad representativeness.

### E.4    COMPLETE EXAMPLES IN ARTIMUSE-10K

In ArtiMuse-10K, professional experts meticulously evaluate each image across eight aesthetic attributes, providing detailed textual assessments along with an overall aesthetics score. Here, we present the complete data examples form each main categories in the dataset, including Photography, Painting & Calligraphy, AIGC, 3D Design and Graphic Design, as shown in Fig. 20, Fig. 21, Fig. 22, Fig. 23, Fig. 24, Fig. 25, and Fig. 26.

### E.5    FURTHER COMPARISON OF TEXTUAL ANALYSIS

We provide comprehensive examples of ArtiMuse's structural aesthetic analysis on images, accompanied by expert commentary and comparative evaluations with other models, as illustrated in Fig. 27, Fig. 28, and Fig. 29. All images used in this analysis are sourced from the ArtiMuse-10K test set.

### E.6    RESULTS ON REAL-WORLD IMAGES

To evaluate ArtiMuse's capability in processing out-of-distribution images, we employed real-world images for testing. As demonstrated in Fig. 30, Fig. 31 and Fig. 32, our model maintains accurate and expert-level analysis even when handling real-world scenarios. The results showcase ArtiMuse's ability to provide professional aesthetic assessments, systematically identifying both strengths and weaknesses based on detailed visual characteristics.

## F    DECLARATION OF USE OF LARGE LANGUAGE MODELS (LLM)

We confirm that this paper was written primarily by the authors. Large Language Models (LLMs) were used only as general-purpose tools for language refinement, including grammar correction and stylistic polishing. In particular, GPT-5 (OpenAI, 2025) was employed for minor rephrasing to improve clarity and readability. No LLM was involved in research ideation, experimental design, data analysis, or generation of substantive content.

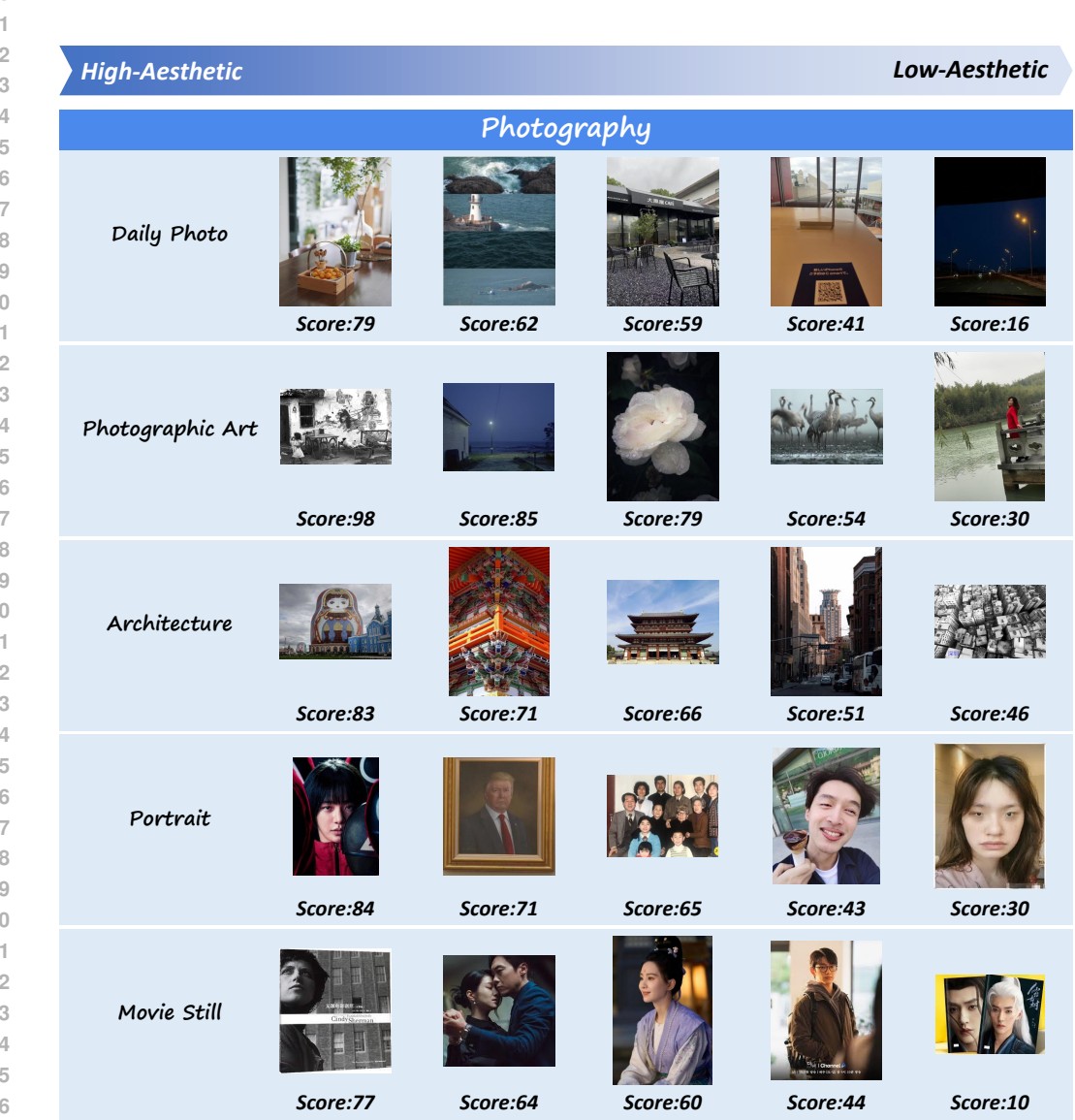

Figure 16: Image examples from the *Photography* category in ArtiMuse-10K dataset.

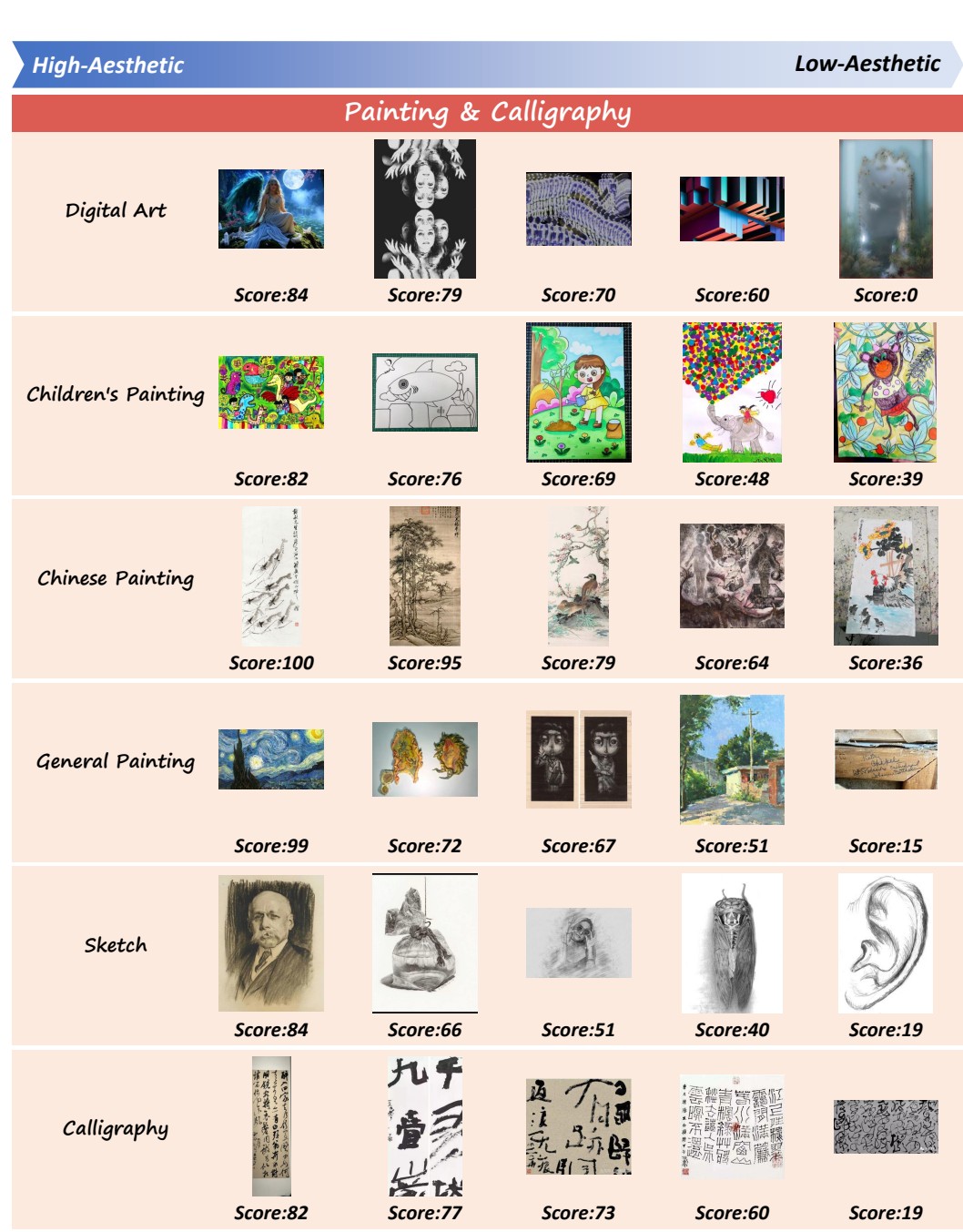

Figure 17: Image examples from the *Painting & Calligraphy* category in ArtiMuse-10K dataset.

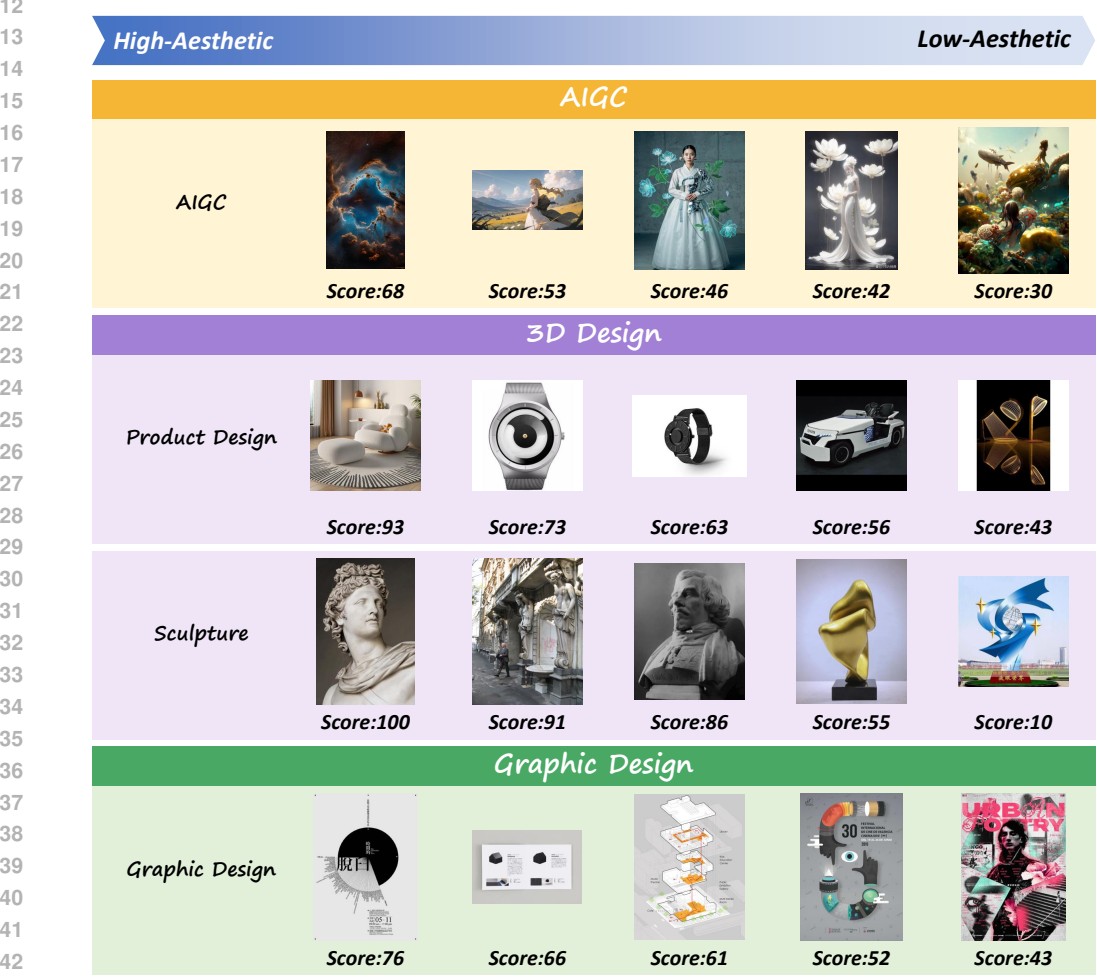

Figure 18: Image examples from the *AIGC*, *3D Design* and *Graphic Design categories* in ArtiMuse-10K dataset.

**High-aesthetic Example**      **Photography**

**1.Composition & Design:**
The composition is standard and effective. The diagonal line of the bridge is used to guide the line of sight. The bridge in the foreground, the traditional buildings in the middle and the modern high-rise buildings in the background are cleverly organized together, forming a rich sense of layering and depth. The reflection on the water surface increases the balance and rhythm of the picture.

**2.Visual Elements & Structure:**
The colors are vivid and contrasting. The warm building lights and the cold night sky and water reflections form a visual impact. The lines and shapes are clear and powerful. The juxtaposition of different architectural styles highlights the characteristics and sense of space of the city.

**3.Technical Execution:**
The exposure is properly controlled, and the rich details of the bright and dark parts are successfully captured in the night scene environment. The focus is accurate, the main body is clear and sharp, and the reflection on the water surface is also well handled. The overall picture quality and detail expression are good.

**4.Originality & Creativity:**
The theme is a common urban night scene. Although the composition and expression techniques are mature, there is no obvious innovative or experimental breakthrough in concept or execution. It is a common and excellent capture of such scenes.

**5.Theme & Communication:**
The theme is clear, vividly conveying the city's unique magical three-dimensional sense and the fusion of the old and the new, and the vibrant night scene. It effectively tells the city's visual story through light and shadow and architectural features.

**6.Emotion & Viewer Response:**
The picture creates a bright, prosperous and vibrant atmosphere, which can effectively arouse the curiosity and admiration of the viewer, making people feel the unique charm of the city and leaving a deeper impression.

**7.Overall Gestalt:**
The overall look is very shocking and attractive, the picture is colorful, rich in details, and clearly layered, and multiple visual elements are harmoniously integrated. The artistry and beauty are outstanding, giving people a complete visual enjoyment, fully showing the charm of the night scene.

**8.Comprehensive Evaluation:**
The photo is highly complete, with standard composition and a sense of hierarchy. The use of colors and visual composition are attractive, the technical execution is solid, and the unique theme and atmosphere of the city are effectively conveyed. The overall visual effect is strong. It is a city landscape photography work with both artistry and beauty.

*Overall Aesthetics Score: 83*

Figure 19: High-aesthetic example from *Photography* category.

**Medium-aesthetic Example**    **Photography**

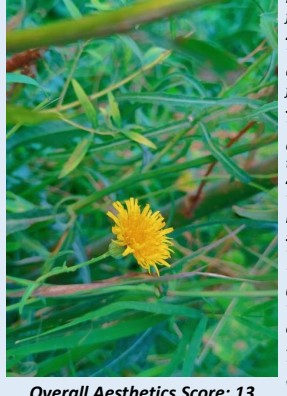

**Overall Aesthetics Score: 52**

**1.Composition & Design:**
Standard composition, using a close-up approach to highlight the plum blossom buds in the foreground, with the background blurred through depth of field. The foreground branches and buds are evenly distributed, but the blurred buildings in the background are slightly distracting, making the overall layout somewhat crowded and lacking stronger visual guiding lines or contrasting elements.
**2.Visual Elements & Structure:**
Vivid color contrast, with the pink buds and blue sky creating a strong visual impact that is quite attractive. The lines of the branches provide a skeletal structure, while the round shapes of the buds add softness. However, the blurred forms in the background feel somewhat disconnected from the sharp details in the foreground.
**3.Technical Execution:**
Technically standard execution. The focus is accurately placed on the foreground buds, achieving a good depth-of-field blur effect. The exposure is appropriate, preserving the vibrant colors of the buds and the purity of the blue sky. Details are acceptable in the focused area, but the blurred background shows noticeable fuzziness.
**4.Originality & Creativity:**
Moderate originality. Close-ups of plum or cherry blossoms are common photography subjects, and the use of depth-of-field blur for the background is also a conventional technique. The inclusion of urban buildings in the background adds a slight contrast between city and nature, but the overall concept and execution do not stand out beyond typical styles.
**5.Theme & Communication:**
The theme is clear, primarily showcasing the budding plum blossoms of spring, conveying vitality, hope, and seasonal change. The narrative is weak, focusing mainly on static beauty. It effectively communicates the essence of spring.
**6.Emotion & Viewer Response:**
Moderate emotional evocation. It allows viewers to sense the beauty and hope of spring but does not elicit deeper emotional resonance. Viewer engagement is average, mostly limited to visual appreciation, lacking a more lasting impression or personally meaningful trigger.
**7.Overall Gestalt:**
The overall impression is decent. The colors and foreground subject are relatively appealing, offering some aesthetic value. As a photographic work, its artistic quality is moderate. The blurred buildings in the background slightly disrupt the harmony, failing to create a stronger or more unique visual impact.
**8.Comprehensive Evaluation:**
Moderate completeness. The close-up composition clearly defines the subject, but the background treatment is less than ideal, diminishing the overall purity. The vibrant colors are a highlight, and the technical execution aligns with mobile photography standards. Creativity lacks uniqueness, and while the theme is clearly communicated, the emotional depth is insufficient. Overall, this is an ordinary yet somewhat visually pleasing spring photography piece.

**Low-aesthetic Example**    **Photography**

**Overall Aesthetics Score: 13**

**1.Composition & Design:**
The composition lacks clear guidance and focus, the main flower is disturbed by the surrounding messy branches and leaves, the background is too busy, the overall layout seems casual, and no effective visual balance or sense of rhythm is formed.
**2.Visual Elements & Structure:**
The color is mainly green, and the yellow flowers provide contrast, but the green part appears to be highly saturated and lacks layering. The lines are mainly messy grass leaves, and no organized or beautiful form and space relationship is formed.
**3.Technical Execution:**
There are obvious deficiencies in technical execution. The picture is not focused, the main flower appears blurred, and although the background is somewhat blurred, the overall clarity is low. The exposure is basically appropriate but fails to make up for the lack of focus.
**4.Originality & Creativity:**
The originality is low. The scene of shooting a small flower is very common. There is no unique creativity or imagination in the perspective, expression method or concept. It belongs to the common casual shooting.
**5.Theme & Communication:**
The theme is clearly about photographing flowers, conveying the presence of small flowers in the natural environment. The narrative or deeper meaning is limited, mainly staying on the surface of the visual presentation.
**6.Emotion & Viewer Response:**
The picture creates a bright, prosperous and vibrant atmosphere, which can effectively arouse the curiosity and admiration of the viewer, making people feel the unique charm of the city and leaving a deeper impression.
**7.Overall Gestalt:**
The emotional arousal is weak, and it is difficult to arouse strong resonance or deep thinking from the viewer. The viewer's participation is not high, the picture lacks fascinating details or artistic conception, and it is difficult to form a lasting impression.
**8.Comprehensive Evaluation:**
The overall look is not good, the picture appears messy and lacks clarity, the artistry is low, and the appeal is insufficient. Technical defects seriously affect the overall visual experience.

Figure 20: Medium-aesthetic example and low-aesthetic example from *Photography* category.

**High-aesthetic Example**                                   **Painting & Calligraphy**

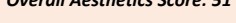

**Overall Aesthetics Score: 95**

**1.Composition & Design:**
The composition is standard, placing the main rabbit to the left of the center of the picture, leaving appropriate space, the overall layout is balanced and stable, and the shape arrangement is natural and sculptural.
**2.Visual Elements & Structure:**
The color is delicately used, and the texture and three-dimensional sense of the rabbit's fur are expressed through the layered changes of brown and gray. The lines are precisely outlined, the shape is captured vividly, and the expression of light and shadow and volume is outstanding.
**3.Technical Execution:**
The mastery of media materials is outstanding. Whether it is the detailed depiction of hair texture or the treatment of details such as eyes and claws, it shows unparalleled techniques and in-depth understanding of nature, and profound realistic skills.
**4.Originality & Creativity:**
Although it is a realistic depiction of animals, such a delicate, accurate and vital naturalistic depiction is highly original and groundbreaking in the context of the times in which it is located, showing a unique observation perspective and experimentality.
**5.Theme & Communication:**
The theme is clear, effectively conveying the rabbit's vivid posture and delicate physiological characteristics, with a certain narrative, showing the artist's awe of nature and meticulous observation.
**6.Emotion & Viewer Response:**
It can arouse the viewer's curiosity and closeness to natural life. The rabbit's eyes and posture are contagious, which arouses attention and appreciation of details, leaving a deep and lasting impression.
**7.Overall Gestalt:**
The overall look is extremely outstanding, with extremely high aesthetics and artistry. It is a model of realistic painting. The overall impression is harmonious and fascinating, showing outstanding artistic achievements.
**8.Comprehensive Evaluation:**
The work is highly complete, with a solid and balanced composition, rich visual elements and exquisite processing, and superb technical execution, showing excellent realism and in-depth observation of nature. The theme is clear, effectively conveying the beauty of life, which can arouse the viewer's emotional resonance and admiration for artistic skills. The overall artistry is extremely high, and it is an outstanding example of naturalistic depiction with important historical and artistic value.

**Medium-aesthetic Example**                                 **Painting & Calligraphy**

**Overall Aesthetics Score: 51**

**1.Composition & Design:**
The composition is standard, with a relatively centered approach, the building as the main body, and walls and plants on both sides, forming a basic balance. The layout is direct and easy to understand, without complex spatial processing, and the focus is on the building facade and its doorway area.
**2.Visual Elements & Structure:**
The use of color is relatively direct, the warm color of the red bricks contrasts with the green of the plants, and the blue sky and white clouds also create a sense of outdoor light. The lines and shapes are relatively simplified, with a strong sense of brushstrokes, and the form and space expression are relatively flat, but the basic outline is clear.
**3.Technical Execution:**
The mastery of media materials is reflected in the use of brushstrokes, with a strong sense of pigment stacking, showing a simple texture. The details are relatively general, focusing on capturing the overall impression rather than detailed depiction, showing a certain degree of painting directness.
**4.Originality & Creativity:**
The originality and creativity are common. The theme is an ordinary rural building scene, and the painting style is also a relatively direct and simple expression method, without obvious uniqueness or experimental attempts.
**5.Theme & Communication:**
The theme is clear and clearly conveys the image of an ordinary building and its surroundings. The communication is effective, and the viewer can directly recognize the content of the picture, showing a life-like scene.
**6.Emotion & Viewer Response:**
The emotional arousal is relatively bland, and the viewer may feel a simple and peaceful atmosphere, but not strong emotions. The viewer participation and lasting impression of the work are relatively ordinary, relying on the viewer's personal association.
**7.Overall Gestalt:**
The overall look is simple and natural, the color combination is harmonious and has a certain visual impact. The beauty is reflected in its simple depiction and the brushstrokes of the painting. The artistry belongs to the basic realistic or impressionistic style, with a certain ornamental value and medium appeal.
**8.Comprehensive Evaluation:**
The completeness is average, the composition is direct and balanced, the color application is relatively bright, and the technical execution is simple and has a sense of brushstrokes. The theme is clearly conveyed and effective, but the originality and creativity are common, and the emotional arousal and viewer response are relatively bland. The overall beauty and appreciation are simple, and it is a painting that sincerely depicts the scenery in front of the eyes.

Figure 21: High-aesthetic and medium-aesthetic example from *Painting & Calligraphy* category.

**Low-aesthetic Example**        **Painting & Calligraphy**

*Overall Aesthetics Score: 21*

**1.Composition & Design:**
The composition is relatively centered, with the main body being the horse's head and part of the shoulder and neck, leaving a lot of white space. The sense of balance is average, lacking significant guide lines or classic composition techniques to enhance the formal beauty of the picture.
**2.Visual Elements & Structure:**
The use of colors and lines is basic, and the form and sense of space are attempted to be expressed through simple line outlines and shadows, but the structural understanding and three-dimensional expression of the horse are insufficient.
**3.Technical Execution:**
The mastery of media materials is preliminary, the brushstrokes are somewhat immature, the shadow processing is not delicate and systematic, and there are obvious traces of smearing. The depiction of details, such as the eyes, nose and bit, is limited in expression.
**4.Originality & Creativity:**
The theme is a common subject in painting practice. The expression technique is also relatively traditional, lacking unique perspectives, concepts or experimental attempts.
**5.Theme & Communication:**
The theme is clear, and it is a portrait of a horse. The general form of the horse is conveyed, but the horse's spirit or richer story is not deeply expressed.
**6.Emotion & Viewer Response:**
The emotional arousal is weak, and the work is more of a basic modeling depiction, which is difficult to arouse strong resonance or lasting impression from the viewer. The viewer's participation may remain at a simple cognition of its techniques and forms.
**7.Overall Gestalt:**
The overall look is like a practice, and the artistry and appeal are relatively bland. The various elements are combined to form a recognizable image of a horse, but lack a deeper overall beauty and artistic tension. The original image is hand-painted, and there is no distortion of facts or physical reality.
**8.Comprehensive Evaluation:**
The completeness is average and the composition is regular. The visual elements and technical execution levels show basic abilities, but the details and the expression of volume and structure need to be strengthened. The creativity and theme are relatively plain, and the emotional resonance is insufficient. Overall, this is a basic modeling exercise.

**High-aesthetic Example**        **AIGC**

*Overall Aesthetics Score: 78*

**1.Composition & Design:**
The composition is standard and balanced, using vertical rocks as a frame, and the horizontal queue of people and the reflection of the water surface form a contrast and visual guidance. The overall layout is stable but not lacking in layering. The rhythm is reflected through the repeated figures, and the focus is on the marching team on the platform.
**2.Visual Elements & Structure:**
The color is soft and unified, mainly low-saturation blue-gray tones. The warm tones of the characters' clothes are the highlights of the picture. The lines are smooth and the shapes are simple. The sense of space is reflected through the blurring of the near-field rocks and the distant view, and the overall visual composition is harmonious.
**3.Technical Execution:**
The overall rendering effect of the AI-generated image is good, and the texture simulation increases the sense of art, but the details of the characters are slightly blurred and lack fineness. Although the reflection processing is effective, it is slightly stiff at the junction with the main body. The overall technical execution has reached a certain level.
**4.Originality & Creativity:**
Originality is reflected in the combination of oriental artistic conception and fantasy elements. The concept of characters moving forward on a suspended platform has a certain imagination, and the style is unique and beyond the common style, bringing a novel visual experience to the viewer.
**5.Theme & Communication:**
The theme is clear, and it seems to convey a mood about journey, exploration or spiritual pursuit. It has a certain narrative, which triggers the viewer's association with the story behind the picture, and effectively creates a mysterious and solemn atmosphere.
**6.Emotion & Viewer Response:**
The picture successfully evokes tranquil, mysterious or slightly melancholy emotions, inviting the viewer to enter this surreal scene, which has a certain appeal. Although the character's face is blurred, his posture and environment together create a thought-provoking artistic conception.
**7.Overall Gestalt:**
The overall look and feel has a high aesthetic and artistic quality, a unified and attractive style, and although it is generated by AIGC, there is no obvious unreasonableness or distortion, and it successfully creates a complete fantasy world with oriental charm.
**8.Comprehensive Evaluation:**
The completeness is high, the composition uses frame and horizontal line processing, the visual elements are harmonious in color, although the technical execution can be improved in details, but the original concept and theme are well conveyed, the emotion is successfully aroused, and the overall artistic style and appeal are unique, with a strong sense of artistic conception.

Figure 22: Low-aesthetic example from *Painting & Calligraphy* category and high-aesthetic example from *AIGC* category.

**Medium-aesthetic Example**

**AIGC**

**1.Composition & Design:**
The composition uses the contrast between the people and vehicles in the foreground and the huge flamingo in the background to guide the eye. The neck of the flamingo forms repeated lines, which brings a certain sense of rhythm. The focus is on the child's expression. Although the overall layout serves the concept of surrealism, it does not use particularly innovative or classic composition techniques.

**2.Visual Elements & Structure:**
The colors are soft and contrasting, and the pink flamingo, green car and light background are harmoniously matched to create a dreamy feeling. The form of objects, such as the soft feathers of the flamingo and the hard lines of the car, form a visual contrast. The treatment of scale and space is exaggerated and impactful, serving the surreal theme.

**3.Technical Execution:**
The details and textures of the image rendering are well performed, especially the texture of the flamingo feathers. The light and shadow effects are also relatively natural. However, the combination of some elements and some details of the characters still reveal traces of AI generation, which is not completely seamless.

**4.Originality & Creativity:**
The surreal combination of a giant flamingo and a child in a car is very unique and imaginative, jumping out of common themes and forms of expression, showing a high level of creativity.

**5.Theme & Communication:**
The theme is clear and conveys a sense of fantasy, surrealism, or childhood imagination. The child's curious expression enhances the narrative of the picture and effectively triggers the viewer's thinking and interpretation.

**6.Emotion & Viewer Response:**
The picture can effectively evoke the viewer's curiosity, surprise and other emotions, which is impressive. Its unusual scene easily attracts the viewer to participate in the interpretation and may inspire the viewer's personal associations about dreams, imagination or surrealism.

**Overall Aesthetics Score: 62**

**7.Overall Gestalt:**
The overall look is unique and visually impactful, and the artistry is reflected in the presentation of its surreal concept. Although the concept is attractive and the execution is relatively perfect, certain characteristics inherent in AI-generated images and the significant distortion of physical reality affect its overall beauty and artistic height evaluation under this scoring system.

**8.Comprehensive Evaluation:**
This artwork excels in originality and imaginative concept, effectively conveying its surreal theme through unique visual elements and harmonious colors. While the composition is well-crafted, the execution shows some AI traces, though not overly distracting. The piece sparks viewer curiosity with its distinctive visual impact, though its artistic depth is somewhat limited by medium constraints and exaggerated distortions of reality.

**Low-aesthetic Example**

**AIGC**

**1.Composition & Design:**
Poor composition, close-up cropping appears cramped, the subject's feet are slightly to the left, the background on the right is blurred and contains some difficult-to-identify elements, resulting in a lack of balance and overall beauty in the picture.

**2.Visual Elements & Structure:**
The color is dull, with brown as the main tone, lacking freshness and layering. The lines and shapes are mainly concentrated on the feet, but the overall details are not sharp enough, the sense of shape is limited, and the sense of space is relatively flat.

**3.Technical Execution:**
Poor technical execution, blurred details and lack of clarity, especially the texture of the feet and the details of the fingers are not handled well, presenting a low-quality visual effect overall.

**4.Originality & Creativity:**
The originality is not high, the theme and expression are relatively common, it is a common foot close-up in medical or health care images, and there is no unique perspective or expression method to enhance creativity.

**5.Theme & Communication:**
The theme is clear, effectively conveying the scenes and behaviors of massage, and the viewer can clearly understand that the image content is about foot care or massage.

**6.Emotion & Viewer Response:**
The emotional arousal is limited, the picture fails to fully show the feelings of the experiencer or the concentration of the masseur, lacks emotional elements that can resonate or engage the viewer, and the overall feeling is bland.

**7.Overall Gestalt:**
The overall impression is mediocre, and there are deficiencies in all aspects. Although the theme is clear, the defects in composition, technology and emotional expression weaken its appeal as a work of art or high-quality image.

**Overall Aesthetics Score: 32**

**8.Comprehensive Evaluation:**
The completeness is average, the composition is cramped, the technical execution needs to be strengthened, and the picture lacks clarity and details. The visual elements are bland and the emotional arousal is insufficient. The theme is clearly conveyed, but the overall originality and artistic appeal are lacking.

Figure 23: Medium-aesthetic example and low-aesthetic example from *AIGC* category.

**High-aesthetic Example**

**3D Design**

*Overall Aesthetics Score: 92*

**1.Composition & Design:**
*The composition standard creates a visual sense of rhythm and contrast by presenting products of different colors and shapes side by side, balancing the elements of the picture, with a prominent main body and clear layout, guiding the viewer's eyes to browse the unique design of each product.*
**2.Visual Elements & Structure:**
*The use of color is the highlight of the work. The bright and harmonious color combination enhances the attractiveness of the product. The lines and shapes are simple and smooth, effectively constructing the modern form of the product. The relationship between form and space is properly handled, and the main body appears three-dimensional and prominent against a clean background.*
**3.Technical Execution:**
*The technical execution is excellent, the focus is accurate, the product details are clear, the exposure is even and accurately restores the color and texture of the product, the lighting is soft, and the volume of the product is effectively shaped. The overall photography level is highly professional.*
**4.Originality & Creativity:**
*Originality is reflected in the high uniqueness of the product design itself. The creativity of photography lies in maximizing the innovation of product form and color through minimalist background and precise arrangement. This execution method of focusing on the main creativity makes the image itself have a strong visual freshness.*
**5.Theme & Communication:**
*The theme is clear, effectively conveying the design features, color diversity and modern style of the product, clearly showing the unique structure of each product, and high communication efficiency, allowing viewers to quickly understand the product concept and selling points.*
**6.Emotion & Viewer Response:**
*Color and form evoke the viewer's positive emotions, feel the fashion, vitality and fun of the product, and arouse the viewer's curiosity and interest in the product. The clean presentation method helps the viewer focus on the product itself and generate associations with home or space.*
**7.Overall Gestalt:**
*The overall look is very harmonious and beautiful. All elements work together to create a simple, modern and high-quality artistry. The image is very attractive and successfully conveys the aesthetic value of the product.*
**8.Comprehensive Evaluation:**
*The image excels in completeness, professional composition, and technical execution, effectively highlighting the product's design. The attractive color palette enhances its overall quality. While originality stems from the product design, the presentation brilliantly conveys this creativity. The theme is direct and impactful, evoking positive emotions. A high-standard work with strong commercial and aesthetic appeal.*

**Medium-aesthetic Example**

**3D Design**

*Overall Aesthetics Score: 60*

**1.Composition & Design:**
*The watch is positioned center-left, with the curved strap naturally guiding focus to it. The plain white background ensures clarity but results in a somewhat basic composition—balanced yet lacking creative dynamism.*
**2.Visual Elements & Structure:**
*The watch's metallic case sharply contrasts with its beige, orange, and blue striped strap, reflecting a vibrant, energetic aesthetic. Its clean lines, square dial with round crown, and curved strap design stand out effectively against the white background, though the composition could benefit from more depth.*
**3.Technical Execution:**
*In terms of the mastery of media materials, the image clearly showcases the watch, with accurate details like the strap's fabric texture and dial display. Lighting is even, exposure is correct, and textures/buttons are well-defined. While technically proficient for commercial photography, it lacks exceptional or distinctive techniques.*
**4.Originality & Creativity:**
*In terms of originality and creativity, this is a typical product display photo. The shooting techniques and presentation methods are relatively common, in line with industry standards, and are intended to clearly show the appearance of the product. There is no unique perspective or experimentality in the concept or execution, and the imagination is limited. It is a standardized commercial photography work.*
**5.Theme & Communication:**
*The theme is clear and clearly conveys the appearance and main features of the product, that is, a smart watch. The design sense of the product and the matching special strap are effectively conveyed. As a product display photo, it is not narrative or story-telling, and mainly focuses on the intuitive presentation of functions and appearance, which can effectively tell the viewer what the product looks like.*
**6.Emotion & Viewer Response:**
*The image focuses on product presentation rather than emotional appeal. Viewer engagement relies on interest in the watch's design and features, with lasting impact depending on brand perception. It serves primarily as functional product documentation.*
**7.Overall Gestalt:**
*The image is clean and well-composed, highlighting the product effectively. Its appeal lies primarily in the watch's design and colors, while the photography focuses on accurate, attractive presentation. As a commercial shot, it succeeds as a competent product display without visual distortion.*
**8.Comprehensive Evaluation:**
*This product shot demonstrates strong technical execution with clear, harmonious composition and on-brand color scheme. While creatively conventional, it effectively showcases the watch's details with commercial precision—clean, functional, and visually balanced.*

Figure 24: High-aesthetic and medium-aesthetic example from *3D Design* category.

**Low-aesthetic Example**

**3D Design**

**Overall Aesthetics Score: 28**

**1.Composition & Design:**
The composition standard places the main tire in the center of the picture. The perspective angle can still show its structural characteristics. The overall layout is balanced, but it lacks unique visual guidance or dynamic sense.

**2.Visual Elements & Structure:**
The color is mainly grayscale, functionality is stronger than beauty, the shape and lines clearly depict the tire and its unique tread structure, the repeated lines create a visual rhythm, and the sense of space is presented through the basic 3D grid background.

**3.Technical Execution:**
In terms of the mastery of media materials, the rendering effect is relatively basic, lacking material details and advanced light and shadow performance, the detail processing needs to be strengthened, the edge jagged feeling is obvious, and the modeling grid lines of the background affect the final presentation effect.

**4.Originality & Creativity:**
The originality is mainly reflected in the unique tread design. The concept is somewhat experimental, but the overall expression is similar to the common design display method. The imagination is reflected in the alternative thinking of the tire function, but the overall creativity is limited by its nature as a design draft.

**5.Theme & Communication:**
The theme is clear, effectively conveying the design concept of the tire and its special tread, and clearly showing its morphological structure, but lacking narrative or deeper symbolic meaning.

**6.Emotion & Viewer Response:**
The emotional arousal is limited, the nature of the image is more inclined to technical and design display, and the viewer's participation may be limited to interest in the design itself, and it is difficult to produce a deep or lasting impression.

**7.Overall Gestalt:**
The overall look is closer to a design sketch or technical demonstration, lacking beauty and low artistry. The main attraction lies in its unique design concept, but the limitations of technical presentation seriously affect the overall impression.

**8.Comprehensive Evaluation:**
The completeness is average, the composition highlights the subject in a standard way, the visual elements present a unique design form but the color is monotonous, the technical execution is still in the early stages, the rendering effect is not good and affects the overall perception, the originality is reflected in the design concept rather than the form of expression, the theme is conveyed directly but lacks depth, and the emotional arousal is limited. Overall, this is a technical image that shows the design concept, and the artistry needs to be improved.

**High-aesthetic Example**

**Graphic Design**

**Overall Aesthetics Score: 93**

**1.Composition & Design:**
The composition is balanced, with a blurred wheelchair in the foreground adding narrative depth and guiding attention to the two women supporting each other on the beach. The off-center placement of the figures, combined with the spacious background and blurred foreground, creates layered depth. Red lines enhance visual dynamism, seamlessly connecting the characters and text for a harmonious design.

**2.Visual Elements & Structure:**
The warm, emotional palette contrasts the sandy background with the blue water, while the characters' clothing harmonizes with the environment. The texture blends painterly softness with photographic realism. Smooth lines, particularly the red strokes symbolizing emotional connection, enhance the artwork's expressiveness. The thoughtful handling of form and space accentuates the scene's atmosphere.

**3.Technical Execution:**
The image demonstrates strong technical execution, with natural background blur and foreground depth of field emphasizing the main subjects. Soft lighting enhances character details, while the text integrates seamlessly for clear, unobtrusive messaging. Professionally handled with balanced contrast, it meets high standards for poster design.

**4.Originality & Creativity:**
The poster stands out with creative brilliance, blending beachside figures, blurred foreground elements, and hand-painted text/lines into a poetic, emotive visual language. The innovative red line serves as both decoration and emotional symbolism, elevating the design beyond conventional movie posters with striking artistry and distinctiveness.

**5.Theme & Communication:**
The poster powerfully conveys themes of family, love, and companionship through its central duo, seaside setting, and bold text like "Mom!" The warm tones and characters' intimate posture evoke emotional resonance, while strong visual storytelling hints at their relationship and the film's core message.

**6.Emotion & Viewer Response:**
The tender embrace and held hands between characters radiate warmth and familial love, while the foreground wheelchair adds emotional depth—hinting at themes of care, resilience, and life's bonds. Striking and evocative, the imagery lingers in memory, compelling viewers to discover the film's story.

**7.Overall Gestalt:**
The poster achieves remarkable artistic harmony, blending composition, color, and typography into a uniquely evocative aesthetic. Its emotional depth, thematic resonance, and visual polish create a compelling, authentic impression.

**8.Comprehensive Evaluation:**
This movie poster excels in composition, creativity, and emotional impact. Its balanced design, original concept, and strong thematic clarity create powerful viewer engagement. A professionally executed and highly artistic promotional piece.

Figure 25: Low-aesthetic example from *3D Design* category and high-aesthetic example from *Graphic Design* category.

**Medium-aesthetic Example**

**Graphic Design**

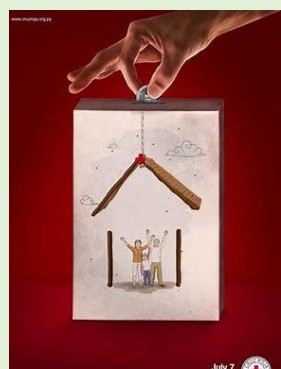

**Overall Aesthetics Score: 52**

**1.Composition & Design:**
The composition is standard, with the central composition highlighting the act of donating and the donation box. The layout of the elements clearly guides the eye, but the overall design lacks a more dynamic or complex sense, the background is simple, and the composition is relatively static.
**2.Visual Elements & Structure:**
The image blends realistic photography with hand-drawn illustrations, creating a contrast between literal action and symbolic meaning. A bold red background emphasizes the theme, starkly contrasting the muted donation box. Clean lines and simple shapes ensure immediate clarity, while the uncluttered composition delivers the message effectively.
**3.Technical Execution:**
In terms of technical execution, the lighting and focus of the photography part are acceptable. The style of the hand-drawn part is deliberately simple, which is in line with the theme expression. The combination of the two media is handled in a relatively basic way, lacking a sophisticated fusion or advanced post-processing technology. In terms of detail processing, the overall tendency is conceptual expression rather than realistic precision.
**4.Originality & Creativity:**
The originality is reflected in the transformation of the donation box into a symbol of home, and the relationship between donation and building a home is presented in a concrete way, with unique creativity and rich imagination.
**5.Theme & Communication:**
The image powerfully conveys donation support for families/shelter through intuitive symbols and text. It visually narrates both the act of giving and its positive impact.
**6.Emotion & Viewer Response:**
It can arouse the viewer's sympathy and positive emotions of helping others. By depicting the assisted family and its symbolic home, it effectively encourages the viewer to have emotional resonance, understand the potential value of their donation behavior, and encourage the viewer to participate.
**7.Overall Gestalt:**
The overall look is good, the creative concept is prominent and the communication is clear. The creativity of combining the donation box with the symbol of the family makes it attractive and highly functional as a public welfare promotional product. The overall presentation presents a kind of beauty that serves the theme expression. Although it is not an extreme work of art, it has good completeness and effective visual narrative. The image is not unreasonably distorted.
**8.Comprehensive Evaluation:**
The completeness is average, the composition is standard but lacks dynamics. The visual elements combine different media to form a conceptual contrast. The creativity is unique, and the abstract donation behavior is concretized into building a home. The theme is clearly conveyed and effectively, with strong narrative, which can arouse the emotional resonance of the viewer. The overall look and feel serves the purpose of publicity and has good appeal and functionality.

**Low-aesthetic Example**

**Graphic Design**

**Overall Aesthetics Score: 34**

**1.Composition & Design:**
The composition uses a center-down approach to highlight the porridge bowl, with oblique tableware and text to form a certain guide line. The layout is acceptable, but the density and balance between elements need to be improved, the overall feeling is a bit crowded, and there is a lack of more impactful or distinctive composition design.
**2.Visual Elements & Structure:**
The colors are mainly dark, and contrasting colors are used to attract attention. The lines and shapes have a hand-painted feel and rich textures. The form and sense of space are expressed through superposition and shadows, but the depth and layering are not prominent enough, and the visual composition is relatively plain.
**3.Technical Execution:**
The mastery of media materials is reflected in the hand-painted texture and brushstrokes, which has a certain stylization. The details such as the expression of the ingredients in the porridge are acceptable, but the overall accuracy and light and shadow effects are not fine enough, and some edge processing is slightly rough.
**4.Originality & Creativity:**
The creativity is reflected in the combination of traditional festival food and illustrations for promotion, but the form and style of expression are relatively common, lacking unique visual concepts or novel expression methods, and the innovation is mediocre.
**5.Theme & Communication:**
The theme is clear, it is about the promotion of Laba porridge. The communication mainly relies on text information. Although the visual part depicts porridge, it lacks in appeal and storytelling. The organization of visual elements fails to effectively strengthen the theme or create a stronger festive atmosphere, and the communication effect is average.
**6.Emotion & Viewer Response:**
The picture attempts to evoke the viewer's intimacy with Laba porridge, but the color and atmosphere are dark, and the emotional arousal is not strong enough. The viewer's participation is low, the picture lacks highlights that attract people's attention or produce lasting impressions, and the generation of personal meaning is also relatively limited.
**7.Overall Gestalt:**
The overall look is acceptable, with a certain hand-painted style. The beauty and artistry are limited by the composition, color and technical execution, and the appeal is insufficient. The overall visual impression is ordinary and lacks highlights.
**8.Comprehensive Evaluation:**
The completeness is average, the composition is acceptable but slightly crowded, the use of visual elements and technical execution are standard. The originality and creativity are limited, and the theme is conveyed by text. The emotional arousal and overall appeal are insufficient, and the overall visual impression is ordinary and lacks outstanding highlights.

Figure 26: Medium-aesthetic example and low-aesthetic example from *Graphic Design* category.

**AIGC**

**Comparison on Attribute 3. Visual Elements & Structure**

**Expert:** The composition of visual elements is average, with rich use of colors, but some color combinations are highly saturated and lack subtle changes. The lines and shapes are relatively simple, and the form expression lacks three-dimensionality. The sense of spatial hierarchy is acceptable, but the overall visual expression is insufficient.

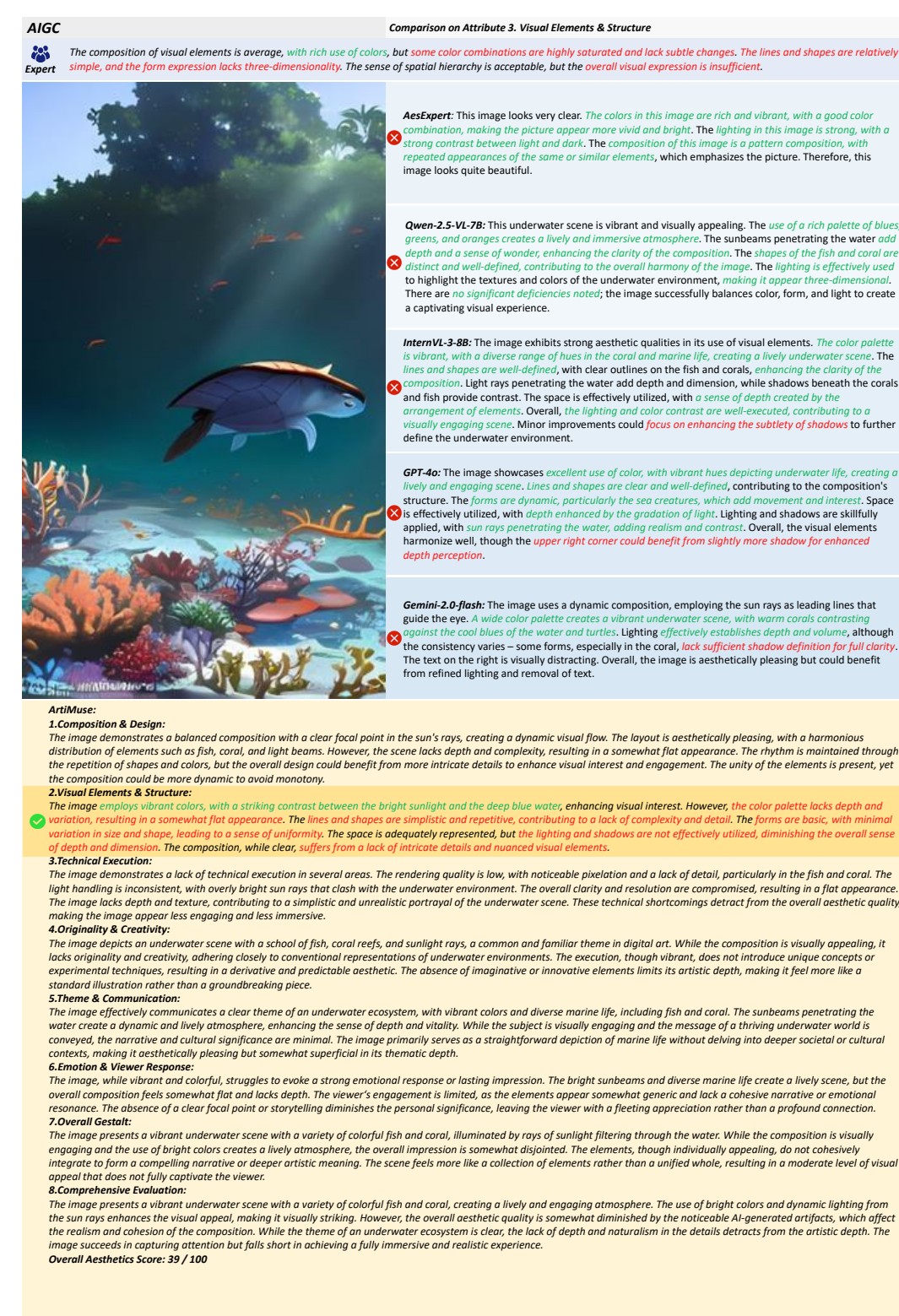

**AesExpert:** This image looks very clear. The colors in this image are rich and vibrant, with a good color combination, making the picture appear more vivid and bright. The lighting in this image is strong, with a strong contrast between light and dark. The composition of this image is a pattern composition, with repeated appearances of the same or similar elements, which emphasizes the picture. Therefore, this image looks quite beautiful.

**Qwen-2.5-VL-7B:** This underwater scene is vibrant and visually appealing. The use of a rich palette of blues, greens, and oranges creates a lively and immersive atmosphere. The sunbeams penetrating the water add depth and a sense of wonder, enhancing the clarity of the composition. The shapes of the fish and coral are distinct and well-defined, contributing to the overall harmony of the image. The lighting is effectively used to highlight the textures and colors of the underwater environment, making it appear three-dimensional. There are no significant deficiencies noted; the image successfully balances color, form, and light to create a captivating visual experience.

**InternVL-3-8B:** The image exhibits strong aesthetic qualities in its use of visual elements. The color palette is vibrant, with a diverse range of hues in the coral and marine life, creating a lively underwater scene. The lines and shapes are well-defined, with clear outlines on the fish and corals, enhancing the clarity of the composition. Light rays penetrating the water add depth and dimension, while shadows beneath the corals and fish provide contrast. The space is effectively utilized, with a sense of depth created by the arrangement of elements. Overall, the lighting and color contrast are well-executed, contributing to a visually engaging scene. Minor improvements could focus on enhancing the subtlety of shadows to further define the underwater environment.

**GPT-4o:** The image showcases excellent use of color, with vibrant hues depicting underwater life, creating a lively and engaging scene. Lines and shapes are clear and well-defined, contributing to the composition's structure. The forms are dynamic, particularly the sea creatures, which add movement and interest. Space is effectively utilized, with depth enhanced by the gradation of light. Lighting and shadows are skillfully applied, with sun rays penetrating the water, adding realism and contrast. Overall, the visual elements harmonize well, though the upper right corner could benefit from slightly more shadow for enhanced depth perception.

**Gemini-2.0-flash:** The image uses a dynamic composition, employing the sun rays as leading lines that guide the eye. A wide color palette creates a vibrant underwater scene, with warm corals contrasting against the cool blues of the water and turtles. Lighting effectively establishes depth and volume, although the consistency varies – some forms, especially in the coral, lack sufficient shadow definition for full clarity. The text on the right is visually distracting. Overall, the image is aesthetically pleasing but could benefit from refined lighting and removal of text.

**ArtiMuse:**

**1. Composition & Design:**
The image demonstrates a balanced composition with a clear focal point in the sun's rays, creating a dynamic visual flow. The layout is aesthetically pleasing, with a harmonious distribution of elements such as fish, coral, and light beams. However, the scene lacks depth and complexity, resulting in a somewhat flat appearance. The rhythm is maintained through the repetition of shapes and colors, but the overall design could benefit from more intricate details to enhance visual interest and engagement. The unity of the elements is present, yet the composition could be more dynamic to avoid monotony.

**2. Visual Elements & Structure:**
The image employs vibrant colors, with a striking contrast between the bright sunlight and the deep blue water, enhancing visual interest. However, the color palette lacks depth and variation, resulting in a somewhat flat appearance. The lines and shapes are simplistic and repetitive, contributing to a lack of complexity and detail. The forms are basic, with minimal variation in size and shape, leading to a sense of uniformity. The space is adequately represented, but the lighting and shadows are not effectively utilized, diminishing the overall sense of depth and dimension. The composition, while clear, suffers from a lack of intricate details and nuanced visual elements.

**3. Technical Execution:**
The image demonstrates a lack of technical execution in several areas. The rendering quality is low, with noticeable pixelation and a lack of detail, particularly in the fish and coral. The light handling is inconsistent, with overly bright sun rays that clash with the underwater environment. The overall clarity and resolution are compromised, resulting in a flat appearance. The image lacks depth and texture, contributing to a simplistic and unrealistic portrayal of the underwater scene. These technical shortcomings detract from the overall aesthetic quality, making the image appear less engaging and less immersive.

**4. Originality & Creativity:**
The image depicts an underwater scene with a school of fish, coral reefs, and sunlight rays, a common and familiar theme in digital art. While the composition is visually appealing, it lacks originality and creativity, adhering closely to conventional representations of underwater environments. The execution, though vibrant, does not introduce unique concepts or experimental techniques, resulting in a derivative and predictable aesthetic. The absence of imaginative or innovative elements limits its artistic depth, making it feel more like a standard illustration rather than a groundbreaking piece.

**5. Theme & Communication:**
The image effectively communicates a clear theme of an underwater ecosystem, with vibrant colors and diverse marine life, including fish and coral. The sunbeams penetrating the water create a dynamic and lively atmosphere, enhancing the sense of depth and vitality. While the subject is visually engaging and the message of a thriving underwater world is conveyed, the narrative and cultural significance are minimal. The image primarily serves as a straightforward depiction of marine life without delving into deeper societal or cultural contexts, making it aesthetically pleasing but somewhat superficial in its thematic depth.

**6. Emotion & Viewer Response:**
The image, while vibrant and colorful, struggles to evoke a strong emotional response or lasting impression. The bright sunbeams and diverse marine life create a lively scene, but the overall composition feels somewhat flat and lacks depth. The viewer's engagement is limited, as the elements appear somewhat generic and lack a cohesive narrative or emotional resonance. The absence of a clear focal point or storytelling diminishes the personal significance, leaving the viewer with a fleeting appreciation rather than a profound connection.

**7. Overall Gestalt:**
The image presents a vibrant underwater scene with a variety of colorful fish and coral, illuminated by rays of sunlight filtering through the water. While the composition is visually engaging and the use of bright colors creates a lively atmosphere, the overall impression is somewhat disjointed. The elements, though individually appealing, do not cohesively integrate to form a compelling narrative or deeper artistic meaning. The scene feels more like a collection of elements rather than a unified whole, resulting in a moderate level of visual appeal that does not fully captivate the viewer.

**8. Comprehensive Evaluation:**
The image presents a vibrant underwater scene with a variety of colorful fish and coral, creating a lively and engaging atmosphere. The use of bright colors and dynamic lighting from the sun rays enhances the visual appeal, making it visually striking. However, the overall aesthetic quality is somewhat diminished by the noticeable AI-generated artifacts, which affect the realism and cohesion of the composition. While the theme of an underwater ecosystem is clear, the lack of depth and naturalism in the details detracts from the artistic depth. The image succeeds in capturing attention but falls short in achieving a fully immersive and realistic experience.

**Overall Aesthetics Score: 39 / 100**

Figure 27: Structural aesthetic analysis results and comparisons on AIGC images from ArtiMuse-10K test set. Red and green denote positive and negative analyses, respectively. ArtiMuse demonstrates superior performance in identifying flaws in low-aesthetic AIGC images, achieving higher accuracy than competing models and closely aligning with expert human evaluations.

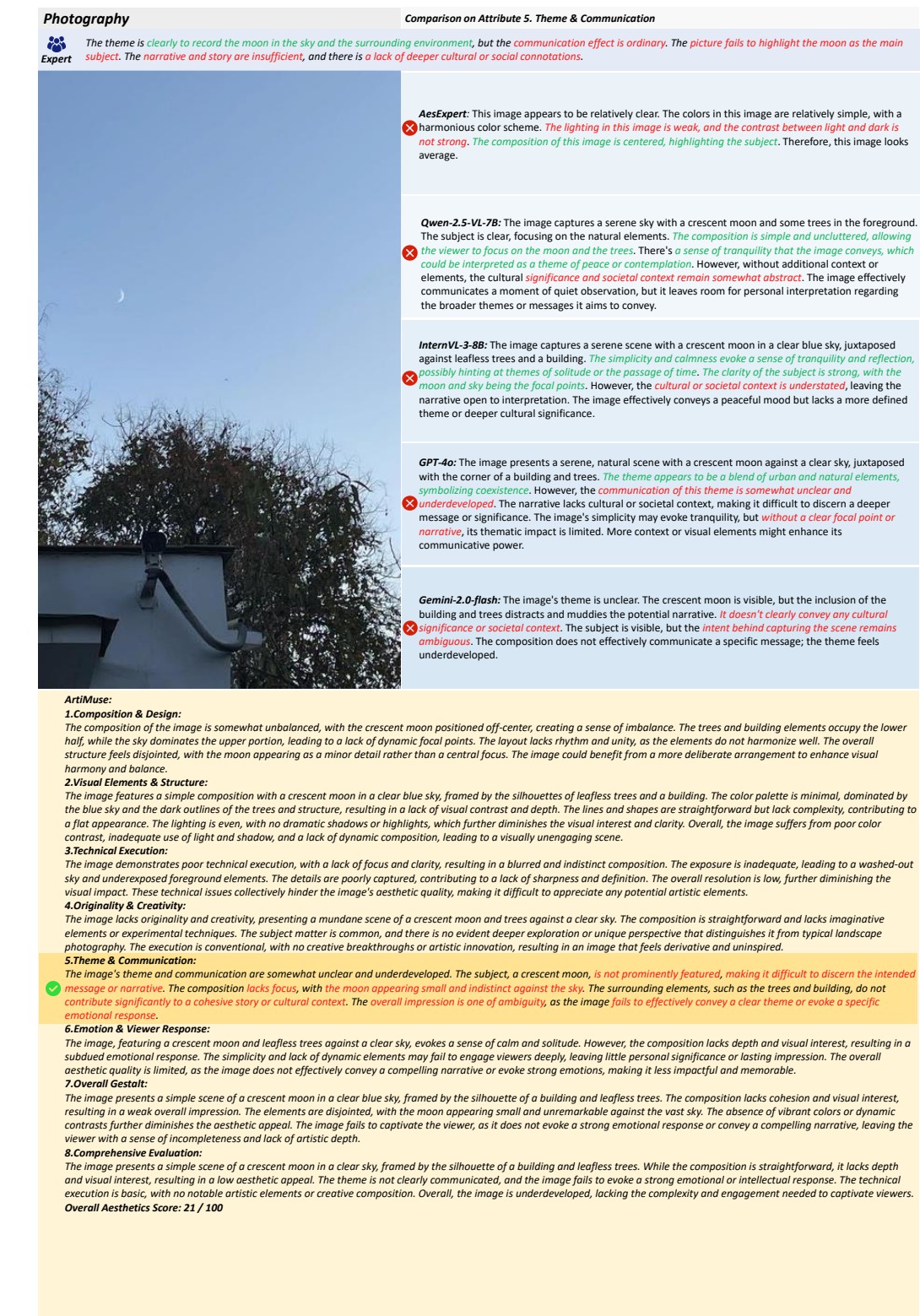

Figure 28: Structural aesthetic analysis results and comparisons on photographic images from ArtiMuse-10K test set. Red and green denote positive and negative analyses, respectively. ArtiMuse demonstrates superior performance in identifying flaws in low-aesthetic photographic images, achieving higher accuracy than competing models and closely aligning with expert human evaluations.

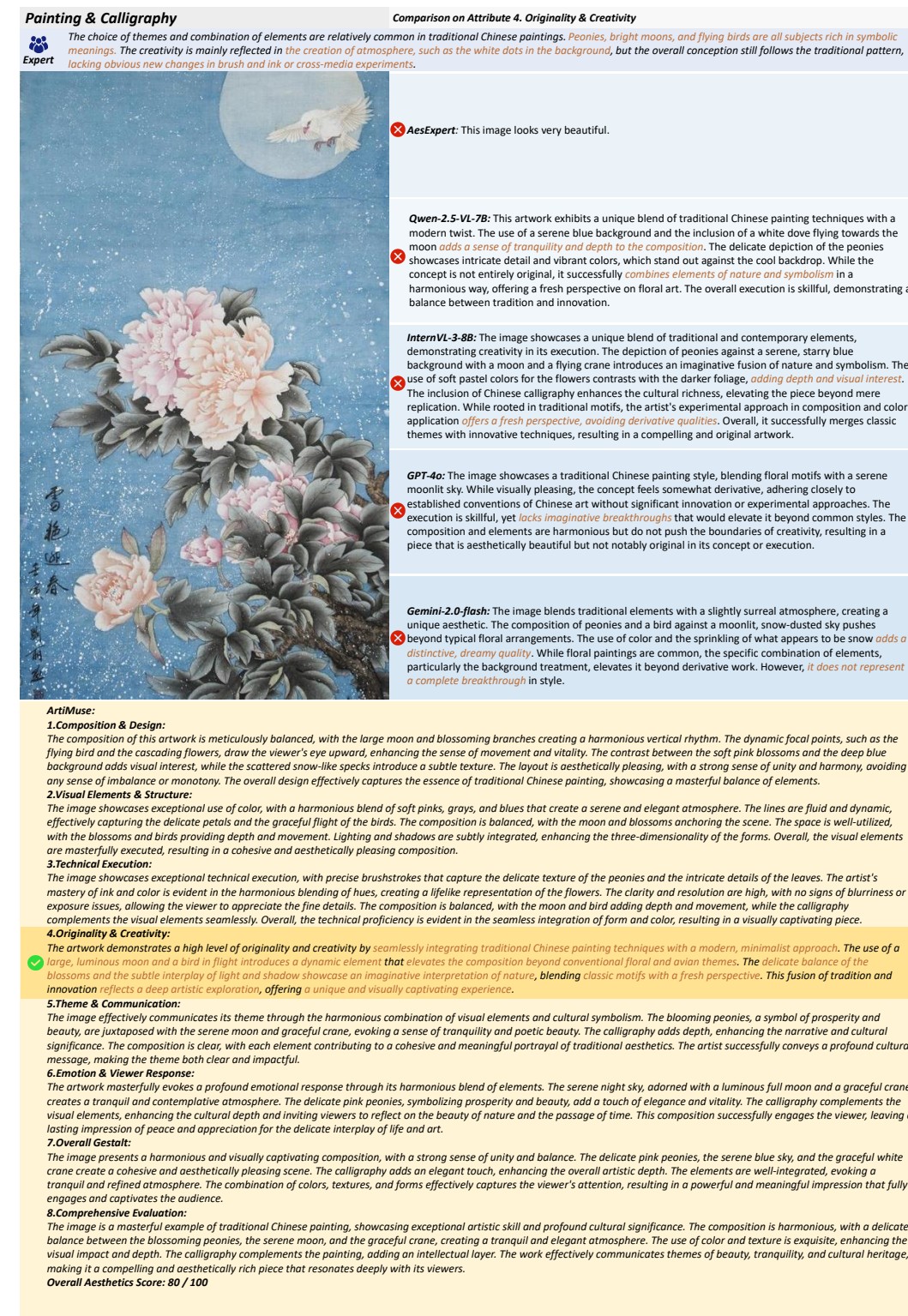

**Painting & Calligraphy**

**Comparison on Attribute 4. Originality & Creativity**

**Expert:** The choice of themes and combination of elements are relatively common in traditional Chinese paintings. *Peonies, bright moons, and flying birds are all subjects rich in symbolic meanings. The creativity is mainly reflected in the creation of atmosphere, such as the white dots in the background,* but the overall conception still follows the traditional pattern, lacking obvious new changes in brush and ink or cross-media experiments.

**AesExpert:** This image looks very beautiful.

**Qwen-2.5-VL-7B:** This artwork exhibits a unique blend of traditional Chinese painting techniques with a modern twist. The use of a serene blue background and the inclusion of a white dove flying towards the moon *adds a sense of tranquility and depth to the composition.* The delicate depiction of the peonies showcases intricate detail and vibrant colors, which stand out against the cool backdrop. While the concept is not entirely original, it successfully *combines elements of nature and symbolism* in a harmonious way, offering a fresh perspective on floral art. The overall execution is skillful, demonstrating a balance between tradition and innovation.

**InternVL-3-8B:** The image showcases a unique blend of traditional and contemporary elements, demonstrating creativity in its execution. The depiction of peonies against a serene, starry blue background with a moon and a flying crane introduces an imaginative fusion of nature and symbolism. The use of soft pastel colors for the flowers contrasts with the darker foliage, *adding depth and visual interest.* The inclusion of Chinese calligraphy enhances the cultural richness, elevating the piece beyond mere replication. While rooted in traditional motifs, the artist's experimental approach in composition and color application *offers a fresh perspective, avoiding derivative qualities.* Overall, it successfully merges classic themes with innovative techniques, resulting in a compelling and original artwork.

**GPT-4o:** The image showcases a traditional Chinese painting style, blending floral motifs with a serene moonlit sky. While visually pleasing, the concept feels somewhat derivative, adhering closely to established conventions of Chinese art without significant innovation or experimental approaches. The execution is skillful, yet *lacks imaginative breakthroughs* that would elevate it beyond common styles. The composition and elements are harmonious but do not push the boundaries of creativity, resulting in a piece that is aesthetically beautiful but not notably original in its concept or execution.

**Gemini-2.0-flash:** The image blends traditional elements with a slightly surreal atmosphere, creating a unique aesthetic. The composition of peonies and a bird against a moonlit, snow-dusted sky pushes beyond typical floral arrangements. The use of color and the sprinkling of what appears to be snow *adds a distinctive, dreamy quality.* While floral paintings are common, the specific combination of elements, particularly the background treatment, elevates it beyond derivative work. However, *it does not represent a complete breakthrough* in style.

**ArtiMuse:**

**1.Composition & Design:**
The composition of this artwork is meticulously balanced, with the large moon and blossoming branches creating a harmonious vertical rhythm. The dynamic focal points, such as the flying bird and the cascading flowers, draw the viewer's eye upward, enhancing the sense of movement and vitality. The contrast between the soft pink blossoms and the deep blue background adds visual interest, while the scattered snow-like specks introduce a subtle texture. The layout is aesthetically pleasing, with a strong sense of unity and harmony, avoiding any sense of imbalance or monotony. The overall design effectively captures the essence of traditional Chinese painting, showcasing a masterful balance of elements.

**2.Visual Elements & Structure:**
The image showcases exceptional use of color, with a harmonious blend of soft pinks, grays, and blues that create a serene and elegant atmosphere. The lines are fluid and dynamic, effectively capturing the delicate petals and the graceful flight of the birds. The composition is balanced, with the moon and blossoms anchoring the scene. The space is well-utilized, with the blossoms and birds providing depth and movement. Lighting and shadows are subtly integrated, enhancing the three-dimensionality of the forms. Overall, the visual elements are masterfully executed, resulting in a cohesive and aesthetically pleasing composition.

**3.Technical Execution:**
The image showcases exceptional technical execution, with precise brushstrokes that capture the delicate texture of the peonies and the intricate details of the leaves. The artist's mastery of ink and color is evident in the harmonious blending of hues, creating a lifelike representation of the flowers. The clarity and resolution are high, with no signs of blurriness or exposure issues, allowing the viewer to appreciate the fine details. The composition is balanced, with the moon and bird adding depth and movement, while the calligraphy complements the visual elements seamlessly. Overall, the technical proficiency is evident in the seamless integration of form and color, resulting in a visually captivating piece.

**4.Originality & Creativity:**
The artwork demonstrates a high level of originality and creativity by *seamlessly integrating traditional Chinese painting techniques with a modern, minimalist approach. The use of a large, luminous moon and a bird in flight introduces a dynamic element that elevates the composition beyond conventional floral and avian themes. The delicate balance of the blossoms and the subtle interplay of light and shadow showcase an imaginative interpretation of nature, blending classic motifs with a fresh perspective. This fusion of tradition and innovation reflects a deep artistic exploration, offering a unique and visually captivating experience.*

**5.Theme & Communication:**
The image effectively communicates its theme through the harmonious combination of visual elements and cultural symbolism. The blooming peonies, a symbol of prosperity and beauty, are juxtaposed with the serene moon and graceful crane, evoking a sense of tranquility and poetic beauty. The calligraphy adds depth, enhancing the narrative and cultural significance. The composition is clear, with each element contributing to a cohesive and meaningful portrayal of traditional aesthetics. The artist successfully conveys a profound cultural message, making the theme both clear and impactful.

**6.Emotion & Viewer Response:**
The artwork masterfully evokes a profound emotional response through its harmonious blend of elements. The serene night sky, adorned with a luminous full moon and a graceful crane, creates a tranquil and contemplative atmosphere. The delicate pink peonies, symbolizing prosperity and beauty, add a touch of elegance and vitality. The calligraphy complements the visual elements, enhancing the cultural depth and inviting viewers to reflect on the beauty of nature and the passage of time. This composition successfully engages the viewer, leaving a lasting impression of peace and appreciation for the delicate interplay of life and art.

**7.Overall Gestalt:**
The image presents a harmonious and visually captivating composition, with a strong sense of unity and balance. The delicate pink peonies, the serene blue sky, and the graceful white crane create a cohesive and aesthetically pleasing scene. The calligraphy adds an elegant touch, enhancing the overall artistic depth. The elements are well-integrated, evoking a tranquil and refined atmosphere. The combination of colors, textures, and forms effectively captures the viewer's attention, resulting in a powerful and meaningful impression that fully engages and captivates the audience.

**8.Comprehensive Evaluation:**
The image is a masterful example of traditional Chinese painting, showcasing exceptional artistic skill and profound cultural significance. The composition is harmonious, with a delicate balance between the blossoming peonies, the serene moon, and the graceful crane, creating a tranquil and elegant atmosphere. The use of color and texture is exquisite, enhancing the visual impact and depth. The calligraphy complements the painting, adding an intellectual layer. The work effectively communicates themes of beauty, tranquility, and cultural heritage, making it a compelling and aesthetically rich piece that resonates deeply with its viewers.
**Overall Aesthetics Score: 80 / 100**

Figure 29: Structural aesthetic analysis results and comparisons on paintings from ArtiMuse-10K test set. Brown denote expert-level analyses. ArtiMuse demonstrates superior technical analysis capabilities for high-aesthetic images, exhibiting higher accuracy than competing models and achieving near-expert-level performance in aesthetic evaluation.

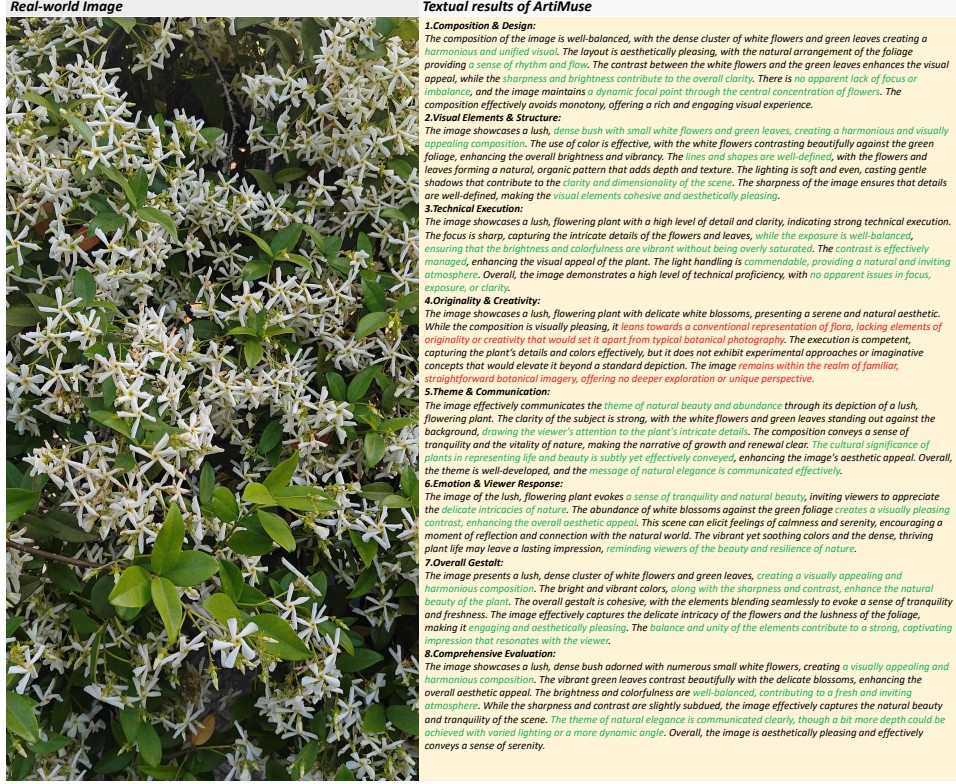

Figure 30: Textual results of ArtiMuse on real-world images. Red and green denote positive and negative analyses, respectively. ArtiMuse delivers expert-level image analysis, offering accurate evaluations of both strengths and weaknesses.

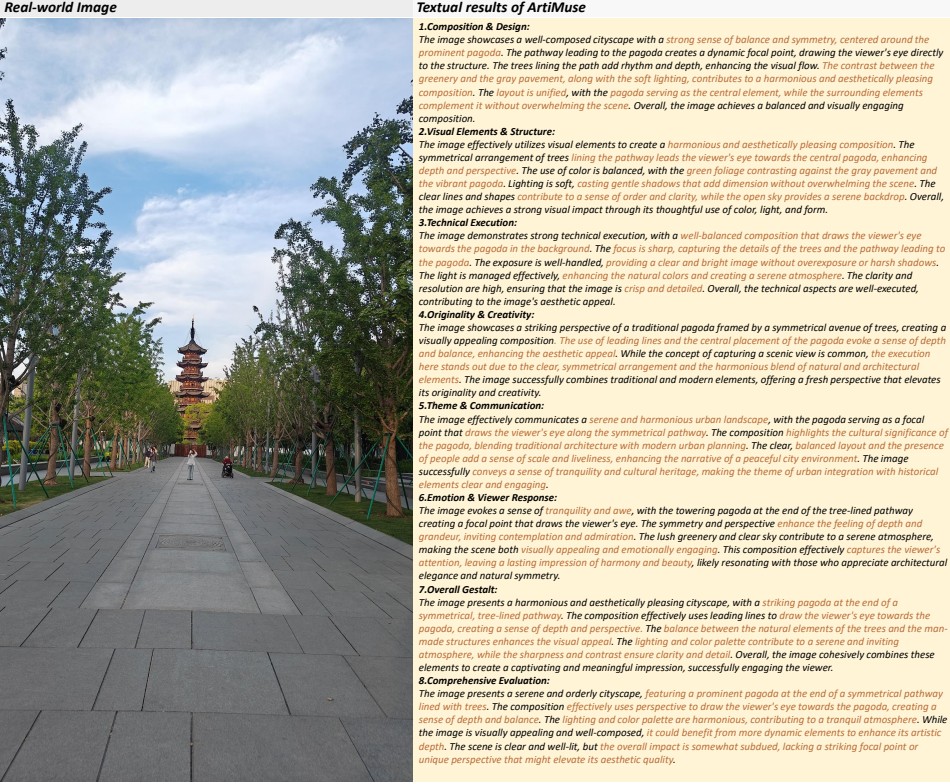

Figure 31: Textual results of ArtiMuse on real-world images. Brown denote expert-level analyses. ArtiMuse is capable of generating expert-level, granular assessments of visual content.

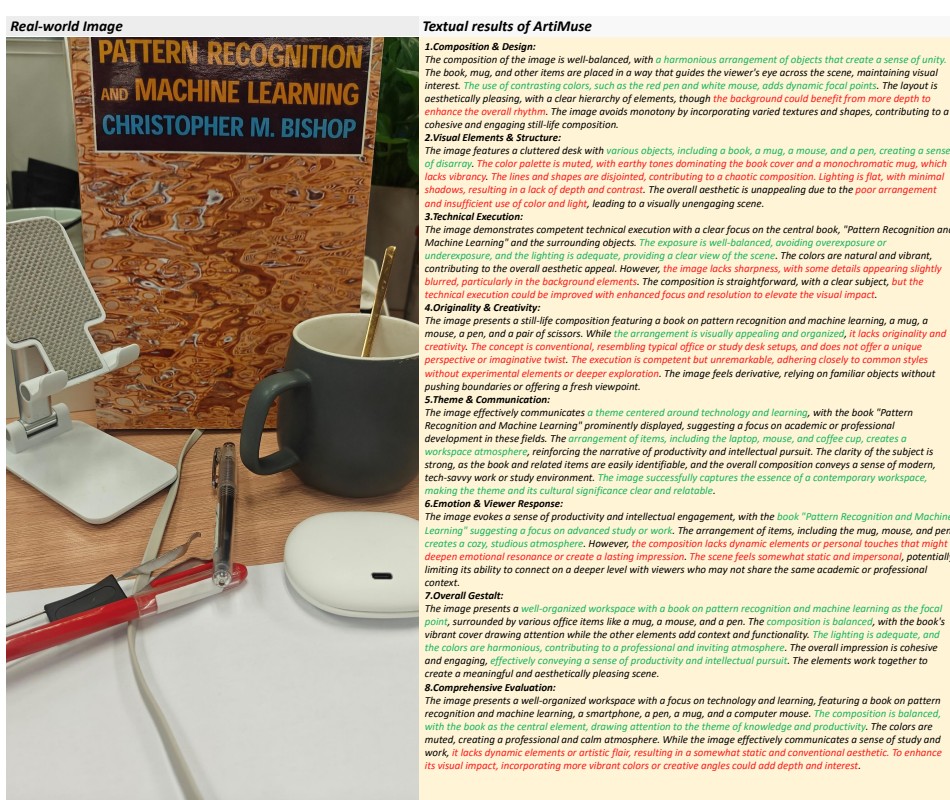

Figure 32: Textual results of ArtiMuse on real-world images. Red and green denote positive and negative analyses, respectively. ArtiMuse delivers expert-level image analysis, offering accurate evaluations of both strengths and weaknesses.

