# OpenReview forum: "ArtiMuse: Fine-Grained Image Aesthetics Assessment with Joint Scoring and Expert-Level Understanding"
_ICLR.cc/2026/Conference — ICLR 2026 Conference Withdrawn Submission_

### Official Review · Reviewer_7pr4 · 2025-10-17

**Soundness:** 3
**Presentation:** 3
**Contribution:** 3
**Rating:** 6
**Confidence:** 4

**Summary:**

This paper proposes ArtiMuse (a multimodal LLM for image aesthetics assessment) and ArtiMuse-10K (10k expert-annotated images with 8 fine-grained analyses + 0–100 scores). ArtiMuse uses two-stage training (text pretraining via LoRA, score finetuning via "Token As Score") and outperforms SOTA models in scoring/text analysis.

**Strengths:**

1. The paper is clearly and coherently written.
2. Its proposed ArtiMuse-10K dataset is meaningful for the IAA field.

**Weaknesses:**

1. In Section 4.2 (lines 273-276), the paper uses a score-guided approach to generate aesthetic captions, which has been used in UniQA [1]. The paper does not properly cite this research.

2. Experiments (a), (b), and (c) should discuss the impact of data size on the results. Images with score annotations may be numerous, so they have a greater impact on the results.

3. I would like to know whether the first stage text training is effective for other methods in the second stage (Text As Score, Level-As-Score). This is related to the generalization performance of the proposed data.

[1] UniQA: Unified vision-language pre-training for image quality and aesthetic assessment

**Questions:**

See above.

---

### Official Review · Reviewer_pXL5 · 2025-10-30

**Soundness:** 3
**Presentation:** 2
**Contribution:** 2
**Rating:** 4
**Confidence:** 3

**Summary:**

This paper proposes a new Multimodal Large Language Model (MLLM) called ArtiMuse for Image Aesthetics Assessment (IAA). The model is capable of simultaneously achieving precise aesthetic scoring and expert-level, fine-grained textual understanding. To realize this, the authors also constructed the ArtiMuse-10K dataset and introduced the Token As Score strategy for continuous score prediction within MLLMs, which are inherently designed for discrete token generation. Experimental results show that ArtiMuse achieved state-of-the-art performance across multiple widely-used benchmarks.

**Strengths:**

1. The expert-annotated dataset (ArtiMuse-10K), featuring professional annotations across many fine-grained attributes (e.g., composition, technical execution, creativity), addresses the issues of coarse granularity, and lack of expert guidance in existing IAA datasets.

2. The ArtiMuse model successfully integrates quantitative scoring with qualitative interpretation.

3. The Token As Score strategy resolves the inherent limitation of MLLMs in performing continuous score prediction by densely mapping existing tokens to a continuous value.

**Weaknesses:**

1.Data Limitations and Distribution: Is a quantity of 10k sufficient, considering the high-dimensional nature of aesthetic assessment? The paper needs to clarify whether the data distribution across different categories is uniform or long-tail, and address the potential risks of data bias.

2.Limited Potential for Foundation Model Enhancement: The fine-tuning process is focused solely on the score prediction task. The paper does not explore or demonstrate the capacity of this task to RL and positively enhance the general capabilities (e.g., reasoning, language understanding) of the foundational MLLM . This limits the perceived value of the fine-tuning beyond the specialized IAA task.

3.Evaluation Reliability: The fairness and reliability of using a Multi-modal Large Language Model (AI) as the sole judge for the quality of aesthetic analysis require stronger theoretical justification. This AI-centric evaluation should be substantiated or replaced by a more comprehensive, large-scale traditional human assessment.

**Questions:**

Answering questions about weaknesses

---

### Official Review · Reviewer_yo8s · 2025-10-31

**Soundness:** 2
**Presentation:** 3
**Contribution:** 2
**Rating:** 4
**Confidence:** 5

**Summary:**

The paper introduces ArtiMuse, a multimodal large language model (MLLM)-based framework for comprehensive Image Aesthetics Assessment (IAA). It integrates joint scoring with expert-level interpretability to provide both holistic and fine-grained aesthetic understanding. Additionally, the authors present ArtiMuse-10K, a professionally curated dataset containing 10,000 images annotated across eight aesthetic dimensions.

**Strengths:**

1.	It introduces an expert-annotated, multidimensional aesthetic dataset (ArtiMuse-10K).
2.	The model, ArtiMuse, provides both holistic scores and fine-grained attribute analysis.
3.	The model enhances interpretability and professional-level understanding.
4.	The model addresses modality bias of prior score-only or text-only models.

**Weaknesses:**

1.	One of the most puzzling aspects of this work is that it mixes different types of images into a single dataset. It remains unclear whether the learned “aesthetic understanding” truly reflects aesthetic principles or merely fits the dataset distribution. For instance, are the scoring standards for Children’s Paintings and Chinese Paintings the same? Although the model performs well during inference without explicit category labels, an additional experiment involving category classification would strengthen the argument.

2.	The paper’s writing could be clearer — particularly regarding the purpose of constructing the 350K dataset and the details of the model training process.

3.	The source of the eight aesthetic attributes is unclear. The appendix states they were specified by “domain experts,” but which specific domain are these experts from, and how authoritative are they?

4.	The basis for category division is ambiguous. Categories such as Children’s Painting and Chinese Painting appear to follow inconsistent classification criteria.

5.	The idea of “Token As Score” is interesting. However, according to the appendix, some selected tokens are numbers (0–9) while others are not. The former inherently preserve ordinal relationships, but the latter do not. Was there any targeted design to handle this inconsistency during training?

6.	In Table 2, the model AesExpert, which underwent aesthetic training, performs worse than models without aesthetic training. Could the authors provide further analysis on this phenomenon?

7.	Regarding existing datasets, how were the rating scales across different datasets standardized or unified?

8.	How was the test set in Table 2 constructed? Was ArtiMuse already trained on the 350K dataset prior to evaluation?

**Questions:**

Please refer to the Weaknesses.

---

### Official Review · Reviewer_FnMT · 2025-11-01

**Soundness:** 3
**Presentation:** 3
**Contribution:** 3
**Rating:** 6
**Confidence:** 3

**Summary:**

This paper presents ArtiMuse, a multimodal large model designed for Image Aesthetic Assessment (IAA), along with the creation of an expert-annotated dataset, ArtiMuse-10K, consisting of 10,000 images annotated across 8 aesthetic attributes. Experimental results show that ArtiMuse outperforms existing methods on multiple benchmarks and achieves significantly higher human preference scores in user studies.

**Strengths:**

1.	This paper introduce a large-scale, expert-labeled dataset that provides valuable resources for advancing research in aesthetic image understanding.
2.	The proposed *Token As Score* strategy is simple yet effective, enabling the model to handle continuous aesthetic scoring naturally.
3.	Extensive experiments across multiple datasets convincingly demonstrate the effectiveness and robustness of ArtiMuse.

**Weaknesses:**

1.	Incomplete review of related work. For example, AesExpert [1] provides a dataset of 21,904 images across three categories with multiple annotated aesthetic attributes.

[1] Huang Y, Sheng X, Yang Z, et al. Aesexpert: Towards multi-modality foundation model for image aesthetics perception[C]//Proceedings of the 32nd ACM International Conference on Multimedia. 2024: 5911-5920.

2.	The rationality of Token selection strategy needs further explanation. The paper proposes using double-letter combinations (e.g., “aa”, “ab”, …) as scoring tokens mapped to a continuous range of 0–100. While this *Token As Score* strategy is innovative, it remains unclear why double-letter combinations were chosen over other alternatives. Why not use numeric tokens, or explicitly defined score tokens instead? Moreover, do these tokens possess any latent semantic meaning in the pretrained vocabulary that could potentially bias the model’s predictions?

3.	The paper claims the model generates expert-level aesthetic commentary, but only presents qualitative examples. Please add quantitative evaluations to systematically assess the generated text quality.

4.	Could you provide an inter-annotator agreement analysis? Given the inherent subjectivity in IAA, it is crucial for verifying the reliability of the dataset.

**Questions:**

Please refer to the Weaknesses section.

---

### Official Review · Reviewer_EX7i · 2025-11-01

**Soundness:** 2
**Presentation:** 3
**Contribution:** 2
**Rating:** 2
**Confidence:** 4

**Summary:**

The paper introduces ArtiMuse, a multimodal large language model for image aesthetics assessment, designed to provide both continuous scores and eight-dimensional textual analysis. It is supported by a new dataset, ArtiMuse-10K, annotated by experts, and a "Token-As-Score" strategy for continuous score prediction. While the work addresses a practical problem and the proposed components demonstrate competitive performance on several benchmarks, its contributions are constrained by significant methodological weaknesses in evaluation, questions about technical novelty, and concerns regarding data integrity and practical applicability, which limit its overall impact.

**Strengths:**

The paper's problem definition is clear and highly relevant, addressing the growing demand for interpretable and quantitative Image Aesthetics Assessment (IAA) in fields like education, content creation, and AIGC quality control. The newly introduced ArtiMuse-10K dataset is a valuable contribution, offering superior coverage and annotation granularity with its five main categories, 15 subcategories, and eight expert-annotated aesthetic attributes, which is a significant improvement over existing resources that offer only scores or general comments. The proposed Token-As-Score strategy is a creative and efficient solution for continuous scoring that cleverly avoids the information loss associated with discretization by leveraging the LLM's existing tokenizer. Furthermore, the pragmatic two-stage training strategy effectively mitigates the common issue where fine-tuning for a scoring task degrades the model's textual generation capabilities.

**Weaknesses:**

Insufficient evaluation rigor
The core claim is expert-level textual analysis, yet the assessment leans heavily on another LLM, Gemini-2.0-flash, as the judge. This setup invites evaluation bias, and the paper does not report how closely the LLM’s judgments align with human experts. Without agreement statistics or confidence intervals, the validity of the text-quality claims remains uncertain. The dataset story is also thin. There is no clear annotation protocol, no inter-rater reliability metrics such as Krippendorff’s alpha, and no description of how disagreements were resolved. These gaps weaken the case that the annotations are dependable.

Limited technical novelty in scoring
Token-As-Score is an appealing idea, but the evidence does not establish clear superiority. Comparisons focus on text-based or discrete-level scoring, while stronger continuous baselines are missing. Readers never see results for a direct numerical regression head or for distribution-based approaches such as Beta regression. Without these head-to-head tests, it is hard to judge the actual innovation and the size of the performance gains.

Risk of data leakage and contamination
Training draws on large public datasets such as AVA, and evaluation is also conducted on these sources. The paper does not document a robust deduplication process or quantify overlap between training and test splits. If near duplicates or exact matches slip through, reported metrics may be inflated, which would blur the picture of true generalization.

Constrained practicality and generalizability
The scoring scheme depends on the token order in the Qwen2.5-7B vocabulary, which complicates portability to other base models and tokenizers. The authors also acknowledge a practical shortcoming. The system explains but does not yet guide improvement. Without actionable suggestions that close the loop from critique to edit, the utility for real creative workflows remains limited.

**Questions:**

To strengthen the work, detailed annotation protocols and inter-rater reliability (IRR) metrics should be reported for the ArtiMuse-10K dataset. The evaluation methodology should be enhanced by expanding the human study component and explicitly reporting the statistical agreement between human and LLM judges. In terms of the model, it is crucial to add experimental comparisons against strong continuous and distribution-based regression baselines to better situate the contribution of the Token-As-Score method. The authors should also conduct a thorough data leakage analysis and investigate the method's portability by testing its adaptation to other LLM families. Finally, future work could explore bridging the gap from analysis to suggestion, thereby creating a more complete and valuable tool for creative applications.

---

### Note · Authors · 2025-11-14

**Comment:**

We respectfully request to withdraw this submission. After internal discussion, our team decide to further extend and reposition the work, and therefore plan to submit an updated version to a more suitable venue. All co-authors have agreed to this withdrawal. We appreciate the consideration of the program committee.

**Withdrawal Confirmation:**

I have read and agree with the venue's withdrawal policy on behalf of myself and my co-authors.